# Direct contribution of skeletal muscle mesenchymal progenitors to bone repair

Anais Julien[1], Anuya Kanagalingam[1], Ester Martínez-Sarrà[1], Jérome Megret[2], Marine Luka[3], Mickaël Ménager[3], Frédéric Relaix ⬡ [1] & Céline Colnot ⬡ [1✉]

Bone regenerates by activation of tissue resident stem/progenitor cells, formation of a fibrous callus followed by deposition of cartilage and bone matrices. Here, we show that mesenchymal progenitors residing in skeletal muscle adjacent to bone mediate the initial fibrotic response to bone injury and also participate in cartilage and bone formation. Combined lineage and single-cell RNA sequencing analyses reveal that skeletal muscle mesenchymal progenitors adopt a fibrogenic fate before they engage in chondrogenesis after fracture. In polytrauma, where bone and skeletal muscle are injured, skeletal muscle mesenchymal progenitors exhibit altered fibrogenesis and chondrogenesis. This leads to impaired bone healing, which is due to accumulation of fibrotic tissue originating from skeletal muscle and can be corrected by the anti-fibrotic agent Imatinib. These results elucidate the central role of skeletal muscle in bone regeneration and provide evidence that skeletal muscle can be targeted to prevent persistent callus fibrosis and improve bone healing after musculoskeletal trauma.

[1] Univ Paris Est Creteil, INSERM, IMRB, Creteil, France. [2] Cytometry core facility, Structure Fédérative de Recherche Necker, INSERM US24/CNRS UMS3633, Paris, France. [3] Laboratory of Inflammatory Responses and Transcriptomic Networks in Diseases, Université de Paris, Imagine Institute, INSERM UMR 1163, Paris, France. ✉email: celine.colnot@inserm.fr

Bone regeneration begins with an inflammatory response, the formation of a fibrous callus and the deposition of cartilage and bone tissues that are then remodeled to reconstitute the initial shape and function of the injured bone. Skeletal stem/progenitor cells activated by the bone injury differentiate into chondrocytes preferentially in the center of the callus where endochondral ossification occurs through the replacement of cartilage by bone[1,2]. At the callus periphery where bone forms via intramembranous ossification, skeletal stem/progenitor cells differentiate directly into osteoblasts. Several sources of skeletal stem/progenitor cells for bone repair have been identified including the bone marrow and the periosteum, which covers the outer surface of bone[3–7]. Other reports have pointed at the contribution of surrounding tissues such as skeletal muscle[8–11]. The role of intact skeletal muscle around bone is essential for bone repair as soft tissue damage is often associated with impaired healing and skeletal muscle coverage can improve bone healing clinically. Yet, the underlying mechanisms are still unknown[12–15]. The presence of bone forming cells in skeletal muscle has been suspected since Urist reported that bone formation could be induced within skeletal muscle[16,17]. Although skeletal muscle stem cells, or satellite cells, have osteochondrogenic potential in vitro and are required for bone repair through their paracrine functions, in vivo investigations revealed a poor osteochondrogenic potential of the myogenic lineage during bone repair[8,11,18]. We sought to explore other cell populations within the skeletal muscle interstitium, containing fibro/adipogenic progenitors (FAP)/mesenchymal progenitors (MP). These cell populations from the skeletal muscle interstitium are known to support skeletal muscle regeneration and become the source of persistent adipose and fibrotic tissues in pathological conditions such as muscular dystrophy[19–23]. In this work, we characterize Prx1-derived skeletal muscle MP participating in cartilage and bone formation during bone repair and investigate their function in a mouse polytrauma model of concomitant injury to bone and adjacent skeletal muscle. Using single-cell RNA sequencing (scRNAseq) analyses of Prx1-derived skeletal muscle MP at d3 and d5 post-injury, we uncover that the initial fibrotic response mediated by skeletal muscle MP and their commitment to the chondrogenic lineage in the fracture callus are impaired following polytrauma. Furthermore, in the polytrauma environment, skeletal muscle surrounding bone is the source of persistent callus fibrosis that can be decreased with Imatinib to improve bone repair.

## Results

**Skeletal muscle participates in bone repair.** To elucidate the tissue origins of skeletal stem/progenitor cells, we co-transplanted Tomato-labeled EDL (*Extensor Digitus Lengus*) muscle and GFP-labeled periosteum grafts at the tibial fracture site of wild type hosts. We observed simultaneous contribution of skeletal muscle-derived and periosteum-derived cells to cartilage and bone within the callus (Fig. 1a). To confirm that the cellular contribution of skeletal muscle was physiological, we transplanted EDL muscle from *GFP-actin* mice in a wild type host and allowed complete regeneration of the transplanted EDL muscle for one month prior to tibial fracture. We observed similar contribution of skeletal muscle-derived cells to cartilage and bone within the fracture callus (Supplementary Fig. 1a). Quantification of GFP+ EDL-derived cells revealed that skeletal muscle-derived cells mainly contribute to cartilage in the center of the callus, indicating specific involvement in the endochondral ossification process (Supplementary Fig. 1b).

To identify the cartilage and bone forming cells derived from skeletal muscle, we first investigated the myogenic lineage by

inducing a tibial fracture in *Pax7^CreERT2;Rosa^mTmG* mice or transplanting EDL muscle from *Pax7^CreERT2;Rosa^mTmG* mice at the fracture site of wild type hosts. GFP+ cells were not detected within the callus demonstrating the lack of physiological contribution of the myogenic lineage (Supplementary Fig. 2). All skeletal stem/progenitor cells forming cartilage and bone in the fracture callus are derived from the Prx1-labeled mesenchymal lineage that marks bone marrow stromal/stem cells and periosteal cells, and is distinct from the Pax3/Pax7-labeled myogenic lineage[5,24]. Transplantation of EDL muscle grafts from *Prx1^Cre;Rosa^mTmG* donors into wild type hosts showed that skeletal muscle also contains MP providing chondrocytes and osteoblasts for bone repair, and strictly derived from the Prx1 mesenchymal lineage (Fig. 1b). The Prx1-derived MP started migrating from skeletal muscle adjacent to bone towards the center of the callus between days 5 and 7 post-fracture and were detected within cartilage from day 7 and within bone until day 21 (Supplementary Fig. 3a). This recruitment of MP from skeletal muscle was triggered by the bone injury since we did not detect GFP+ Prx1-derived cells migrating outside the transplanted muscle in the absence of fracture (Supplementary Fig. 3b). To further confirm the presence of Prx1-derived MP within skeletal muscle, we showed that mononucleated cells isolated from *Prx1^Cre;Rosa^mTmG* hindlimb muscles free of fascia, tendon and fat, and transplanted in wild-type host, were able to integrate into the callus and form cartilage and bone (Supplementary Fig. 3c). Cultured GFP+ Prx1-derived skeletal muscle cells exhibited osteogenic, adipogenic, chondrogenic, and fibrogenic potentials but no myogenic potential, and expressed fibro-mesenchymal, pericyte and tenocyte markers (Supplementary Fig. 3d, e).

**Heterogeneity of skeletal muscle MP.** At steady state, on transverse sections of *Prx1^Cre;Rosa^mTmG* TA (*tibialis anterior*) muscle, Prx1-derived GFP+ cells localized in the skeletal muscle interstitium next to capillaries, co-expressed the pericyte markers NG2 and PDGFRβ, and the mesenchymal markers PDGFRα and CD29 (Fig. 2a). To better understand the composition of the skeletal muscle MP population, we performed scRNAseq analyses of sorted Prx1-derived skeletal muscle cells surrounding the tibia. We identified nine clusters and defined four sub-populations, distinct from myogenic, endothelial and hematopoietic cell populations, and including FAP/MP (expressing *Prrx1*, *Cxcl12*, *Pdgfrα*, *Ly6a*, and *Cd34*), tenocyte-like cells (expressing *Scx*, *Tnmd* and *Kera*), pericytes (expressing *Cspg4*, *Des*, and *Mylk*) and Spp1/Lgals3 population (expressing *Spp1* and *Lgals3*) (Fig. 2b–e and Supplementary Fig. 4a). Flow cytometry analyses showed that freshly isolated Prx1-derived skeletal muscle cells represent 12.5% of mononucleated cells within muscle, are not hematopoietic nor endothelial (CD45−CD11b−CD31− and Tie2−) and coincide with populations expressing the pericyte/mesenchymal marker PDGFRβ, the FAP/MP markers PDGFRα, Sca1, and CD34, but do not encompass all mesenchymal cell types within skeletal muscle (Supplementary Fig. 4b–d). To assess whether Prx1-derived cells represent a unique mesenchymal population within skeletal muscle, we integrated our scRNAseq dataset with published scRNAseq datasets from skeletal muscle. Prx1-derived cells overlapped with FAP/MP (67%), pericytes (14%), tenocyte-like cells (8%), and Spp1/Lgals3 cells (9%) from other datasets, and were distinct from myogenic, endothelial, and immune cells[23,25,26] (Supplementary Fig. 5). Furthermore, we compared the identity of skeletal muscle MP with that of skeletal/stem progenitor cells previously characterized in bone. Chan et al. proposed a definition for skeletal stem/progenitor cells, osteochondroprogenitors and stromal progenitors based on combinations of surface markers in TER119−/CD45−/Tie2−/ITGAV+

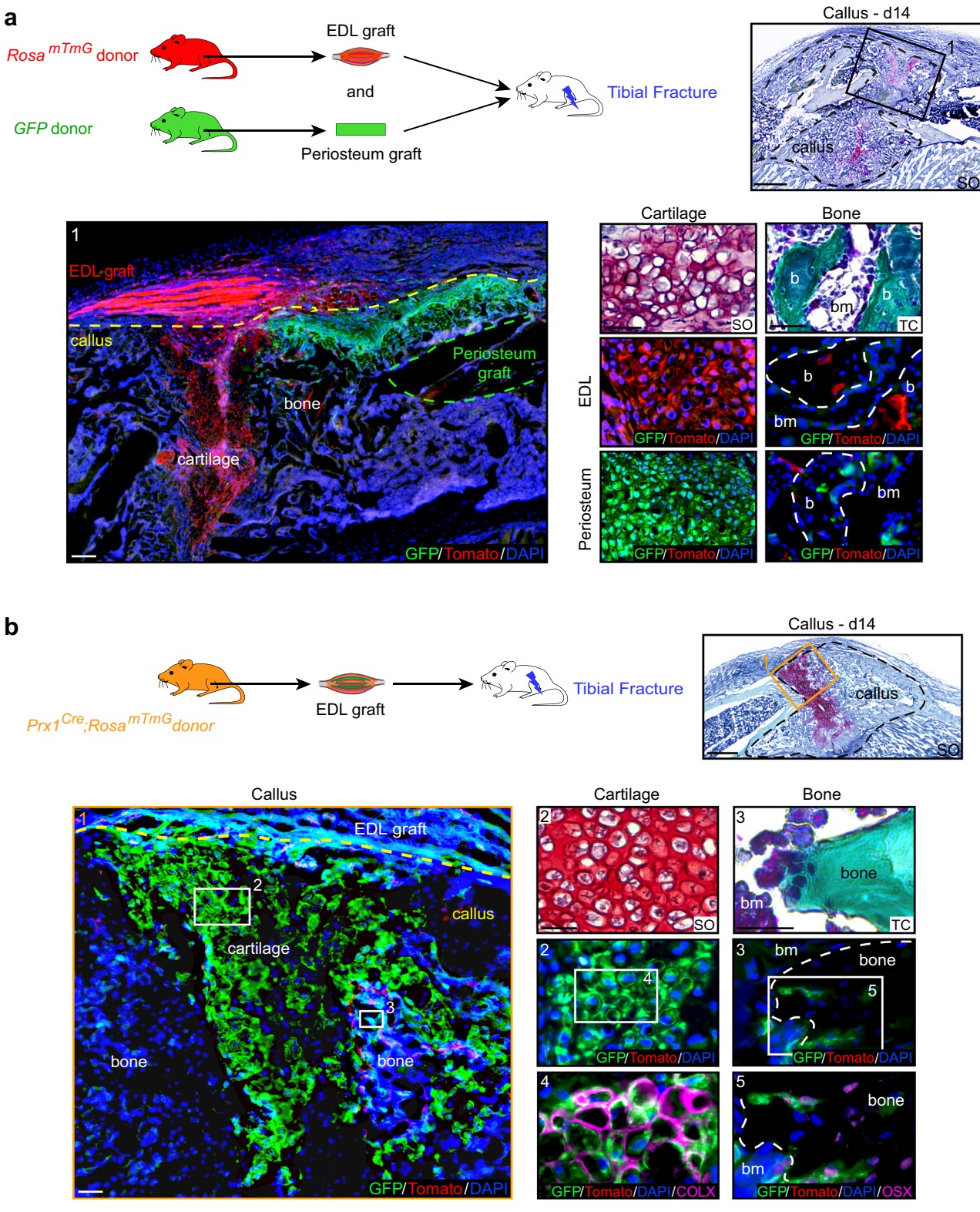

populations within bone[27]. scRNAseq analyses of *Ter119−/Cd45 −/Tie2−/ItgaV+* cells within Prx1-derived skeletal muscle cells suggested the presence of skeletal stem/progenitor cells (*Thy−/6C3−* cells), osteochondroprogenitors (*Thy+/6C3−* cells) and rare stromal progenitors (*Thy−/6C3+* cells) (Supplementary Fig. 4e). In addition, we detected *Ctsk-* and *Cxcl12*-positive cells within FAP/MP cluster while *LepR, Nestin, Mx1* and *Grem1* cells were almost undetectable (Supplementary Fig. 4f)[4,28–32]. Skeletal

muscle thus contains a heterogeneous population of skeletal muscle MP derived from Prx1 mesenchymal lineage, expressing common markers with skeletal stem/progenitor cells from bone and contributing to bone repair.

**scRNAseq analyses of MP after fracture**. To characterize the cellular response of skeletal muscle MP to bone fracture, we

**Fig. 1 Skeletal muscle is a source of chondrocytes and osteoblasts during bone repair. a** Top left, Experimental design of combined grafting of periosteum from *GFP-actin* mice and EDL muscle from *Rosa^mTmG* mice at the site of non-stabilized tibial fracture in wild type hosts. Top right, Longitudinal section of the fracture callus (delimited by a black dotted line) at 14 days post-fracture and stained with Safranin'O (SO). Bottom left, Enlarged view of boxed region 1 on adjacent section counterstained with DAPI. EDL skeletal muscle graft outside the callus and skeletal muscle-derived cells within the callus (Tomato+ signal, callus delimited by a yellow dotted line). Periosteum graft (delimited by a green dotted line) and periosteum-derived cells within the callus (GFP+ signal). Bottom right, High magnification of cartilage and bone (b) derived from the EDL muscle graft (red) or from the periosteum graft (green) stained with SO and Masson's Trichrome (TC) and adjacent sections counterstained with DAPI (bone delimited by a white dotted line). **b** Top left, Experimental design of EDL muscle from *Prx1^Cre;Rosa^mTmG* mice grafted at the site of tibial fracture in wild type hosts. Top right, Longitudinal section of fracture callus (delimited by a black dotted line) at 14 days post-injury and stained with SO. Bottom left, Enlarged view of boxed region 1 on adjacent section counterstained with DAPI. EDL muscle graft outside the callus and skeletal muscle-derived cells within the callus (GFP+ signal, callus delimited by a yellow dotted line). Bottom right, Enlarged view of cartilage (box 2) and bone (box 3) derived from the EDL muscle graft stained with SO and TC. High magnifications of boxed regions 4 and 5 immunostained with COLX and OSX antibodies (magenta, bone delimited by a white dotted line). Scale bar: low magnification = 200 μm, high magnification = 50 μm for cartilage and 25 μm for bone. bm bone marrow. DAPI in blue, GFP in green, Tomato in red. Representative images of three distinct samples.

dissected skeletal muscle surrounding the fracture site, excluding periosteum and bone marrow tissues. After enzymatic digestion, we sorted Prx1-derived GFP+ cells and performed scRNAseq analyses of d0 (uninjured), d3 and d5 post-fracture samples (Fig. 3a). Combined analyses of the three datasets identified 13 clusters in skeletal muscle MP that can be partitioned into four distinct populations (FAP/MP, tenocyte-like cells, pericytes, and Spp1/Lgals3 cells) already defined in un-injured skeletal muscle and a distinct fibroblast cluster (Fig. 3b, Supplementary Fig. 6a, b). This fibroblast cluster was defined as cells expressing genes coding for ECM proteins (*Col1a1*, *Sparc*, *Col3a1*, *Col5a1*, and *Postn*) but no other subpopulation markers (Supplementary Fig. 6c). To assess the fate of skeletal muscle MP in response to fracture, we plotted the mean expression of lineage markers as mesenchymal, fibrogenic (ECM-producing cells), chondrogenic and osteogenic (markers listed in Supplementary information, Table 1). Mesenchymal and fibrogenic markers were expressed mostly by FAP/MP at d0, d3, and d5 post-fracture while chondrogenic markers were only expressed at d5. Osteogenic markers were almost not detected. Detailed analysis per cluster showed that all clusters except Spp1/Lgals3 clusters (clusters 7 and 12) engage in fibrogenic and chondrogenic differentiation, but none engaged in osteogenic differentiation by d5 (Fig. 3c, d and Supplementary Fig. 6d, e). In silico trajectory analysis on d5 post-fracture sample showed that skeletal muscle MP express mesenchymal markers and start expressing fibrogenic markers prior to chondrogenic markers in response to fracture, except for *Sox9* which is already detected in the fibrogenic state (Fig. 3e, f). These results indicate that skeletal muscle MP upon activation adopt a fibrogenic fate before undergoing early chondrogenic differentiation from d5 post-fracture. This cellular response to injury occurs mainly within the FAP/MP population of skeletal muscle MP.

**Musculoskeletal trauma impairs bone healing**. To determine the role of skeletal muscle MP in musculoskeletal trauma, we developed a clinically relevant polytrauma mouse model by inducing mechanical injury to skeletal muscles adjacent to a non-stabilized tibial fracture. As observed in human, musculoskeletal trauma caused fracture non-union, displayed by (i) delayed callus, cartilage, and bone formation, (ii) impaired cartilage and bone resorption, (iii) abnormal fibrous tissue accumulation expressing the fibrotic markers PDGFRα and POSTN, and (iv) absence of bone bridging through day 56 in 100% of cases (Fig. 4a–c and Supplementary Fig. 7a, b). This fracture non-union phenotype was correlated with decreased contribution of skeletal muscle to cartilage (Fig. 4d). The mechanical injury to skeletal muscle alone led to heterogeneous and delayed muscle regeneration as shown by areas containing regenerating muscle fibers and areas

containing fibrous tissue at days 14 and 30 post-injury (Supplementary Fig. 7c). Bone healing was not impaired when combined with TA muscle injury only, thus a threshold of soft tissue trauma exists above which bone healing cannot occur efficiently (Supplementary Fig. 7d). In order to minimize the impact of the mechanical environment in the polytrauma phenotype, we confirmed the findings using a semi-stabilized fracture model. As observed in the non-stabilized fracture model, EDL muscle grafts from GFP-donors contributed to cartilage formation in the fracture callus and skeletal muscle injury led to fibrotic tissue accumulation and absence of bone bridging (Supplementary Fig. 8).

**Trauma alters fibrotic response of MP**. To identify the impact of polytrauma on skeletal muscle MP, we performed integrated scRNAseq analyses of Prx1-derived skeletal muscle cells after fracture and polytrauma (Fig. 5a). Analysis of combined d0, d3 post-fracture, d5 post-fracture, d3 post-polytrauma, and d5 post-polytrauma samples showed that d0, d3, and d5 samples clustered independently of the type of injury (non-stabilized fracture or polytrauma) except for the pericytes and Spp1/Lgals3 clusters (Fig. 5b left). Combined analysis of the five experimental groups uncovered 17 clusters highlighting six populations: FAP/MP, fibroblasts, tenocyte-like cells, pericytes, Spp1/Lgals3 cells, and chondrocytes. Chondrocyte cluster was defined as cells expressing *Col2a1*, *Acan*, and *Fgfr3* (Fig. 5b right, c and Supplementary Fig. 9a, b). Analyses of cell cycle showed no difference in cell proliferation or cell death between fracture and polytrauma, and clusterization with cell cycle regression showed equivalent sub-populations (Fig. 5d, Supplementary Fig. 9c, d). We then determined whether polytrauma impairs the capacity of skeletal muscle MP to engage in fibrogenic and chondrogenic fate. Analyses of mesenchymal and fibrogenic marker expression indicate a delay in down-regulation of mesenchymal markers at d3 post-polytrauma, correlated with a delay in up-regulation of fibrogenic and chondrogenic markers at d3 and d5 post-polytrauma (Fig. 5e). The fibroblast population was detected at d3 and d5 post-fracture, d5 post-polytrauma, and the chondrocyte population at d5 post-fracture (Fig. 5f). We further performed gene ontology (GO) analysis on non-proliferative clusters 2 and 3 at day 3, and cluster 8 at day 5. We used differentially expressed genes between clusters 3 (composed of d3 post-fracture cells) and 2 (composed on d3 post-polytrauma cells), and between post-fracture and post-polytrauma cells within cluster 8 to run GO analysis. In response to fracture, cells showed a high metabolic activity and secreted ECM at d3. In response to polytrauma, cells expressed markers of angiogenesis, hypoxia and immune response but exhibited a lower metabolic activity and lower ECM secretion (Fig. 5g). At d5 post-fracture,

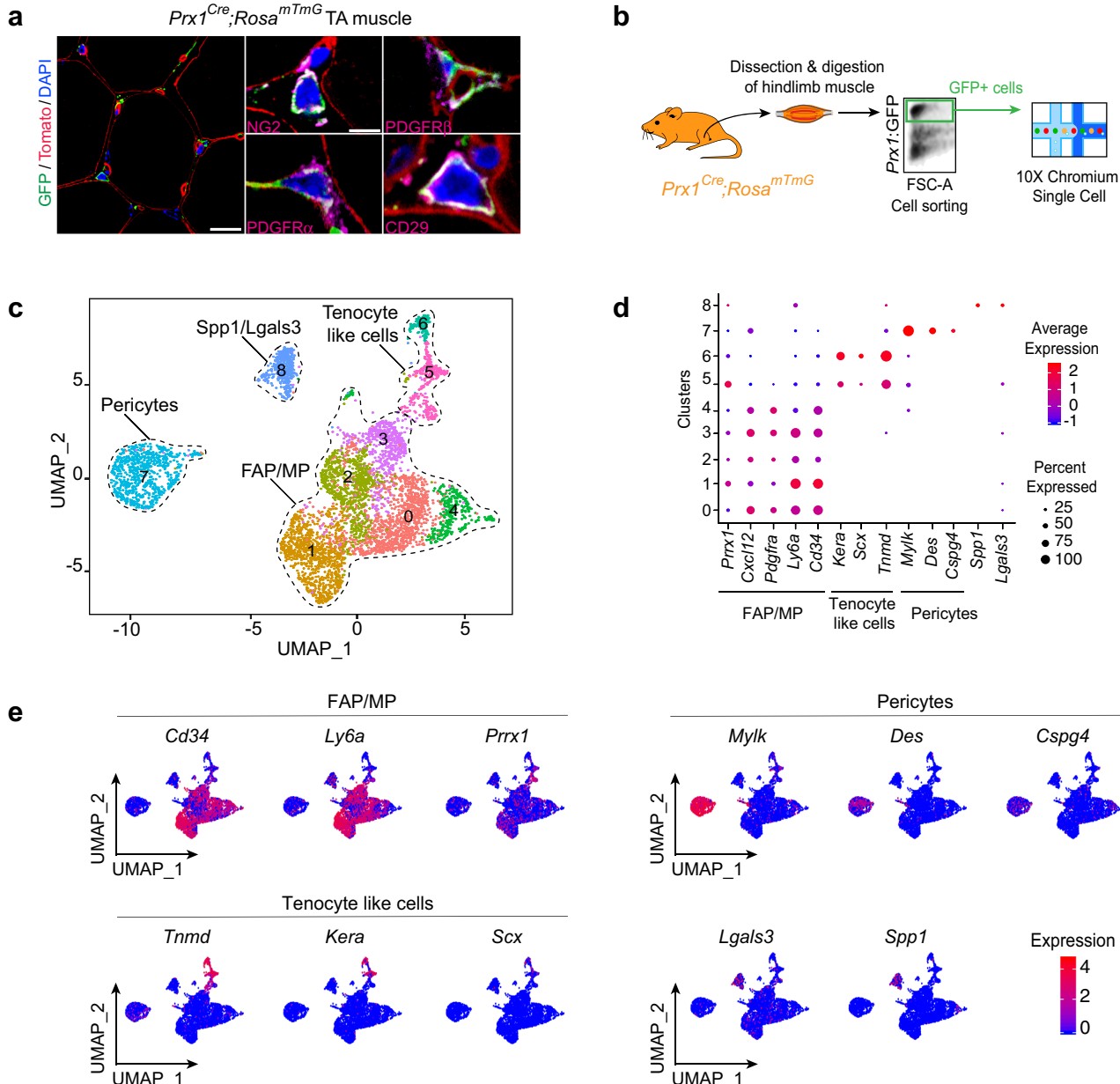

**Fig. 2 Single-cell analyses of skeletal muscle mesenchymal progenitors in intact muscle. a** Transverse sections of TA muscle from *Prx1^Cre;Rosa^mTmG* mice immunostained with NG2, PDGFRβ, PDGFRα, or CD29 antibodies (magenta). DAPI-labeled nuclei in blue, GFP-labeled Prx1-derived cells in green and Tomato labeled non Prx1-derived cells in red. Representative images of three distinct samples. **b** Experimental design of scRNAseq experiment. Prx1-derived skeletal muscle cells were isolated from hindlimb skeletal muscles and sorted based on GFP-expression prior to scRNAseq. **c** UMAP projection of color-coded clustering of Prx1-derived cells reveals nine clusters defining four distinct populations (limited by a black dotted line). **d** Dotplot of indicated gene expression identifying FAP/MP, tenocyte-like cell, pericyte and Spp1/Lgals3 populations. **e** FeaturePlot of sub-population marker expression (*Cd34*, *Ly6a*, and *Prrx1* for FAP/MP, *Tnmd, Kera*, and *Scx* for tenocyte like cells, *Mylk, Des*, and *Cspg4* for pericytes and *Lgals3* and *Spp1* for Spp1/Lgals3 cells). Scale bars: low magnification: 30 μm, high magnification: 10 μm.

cells expressed high level of ECM genes, genes from TGFβ pathway and signaling pathways linked with chondrogenic differentiation (Supplementary information, Table 2). These changes were not observed at d5 post-polytrauma. Instead cells exhibited higher level of metabolic activity suggesting a delay in their activation post-injury (Fig. 5h). These results show that the fibrogenic engagement of skeletal muscle MP is a crucial step during bone repair and precedes chondrogenic differentiation. In polytrauma, skeletal muscle injury perturbs the commitment toward the fibrogenic fate and delays the chondrogenic differentiation of skeletal muscle MP.

**Skeletal muscle is the source of persistent callus fibrosis.** We then analyzed the consequences of polytrauma at later stages of repair. As observed by scRNAseq analysis, the expansion of GFP+ skeletal muscle MP was detected on tissue sections at d3 post-fracture and polytrauma. By day 21 post-injury, GFP+ signal increased around the fracture callus in polytrauma compared to fracture, coinciding with the abnormal accumulation of fibrotic tissue within the callus (Fig. 6a and Fig. 4a, b). Prx1-derived MP gave rise to callus fibrosis (Fig. 6b). Moreover, EDL and periosteum grafting showed that skeletal muscle MP produce bone and fibrous tissue in the callus, while periosteal cells only form bone by

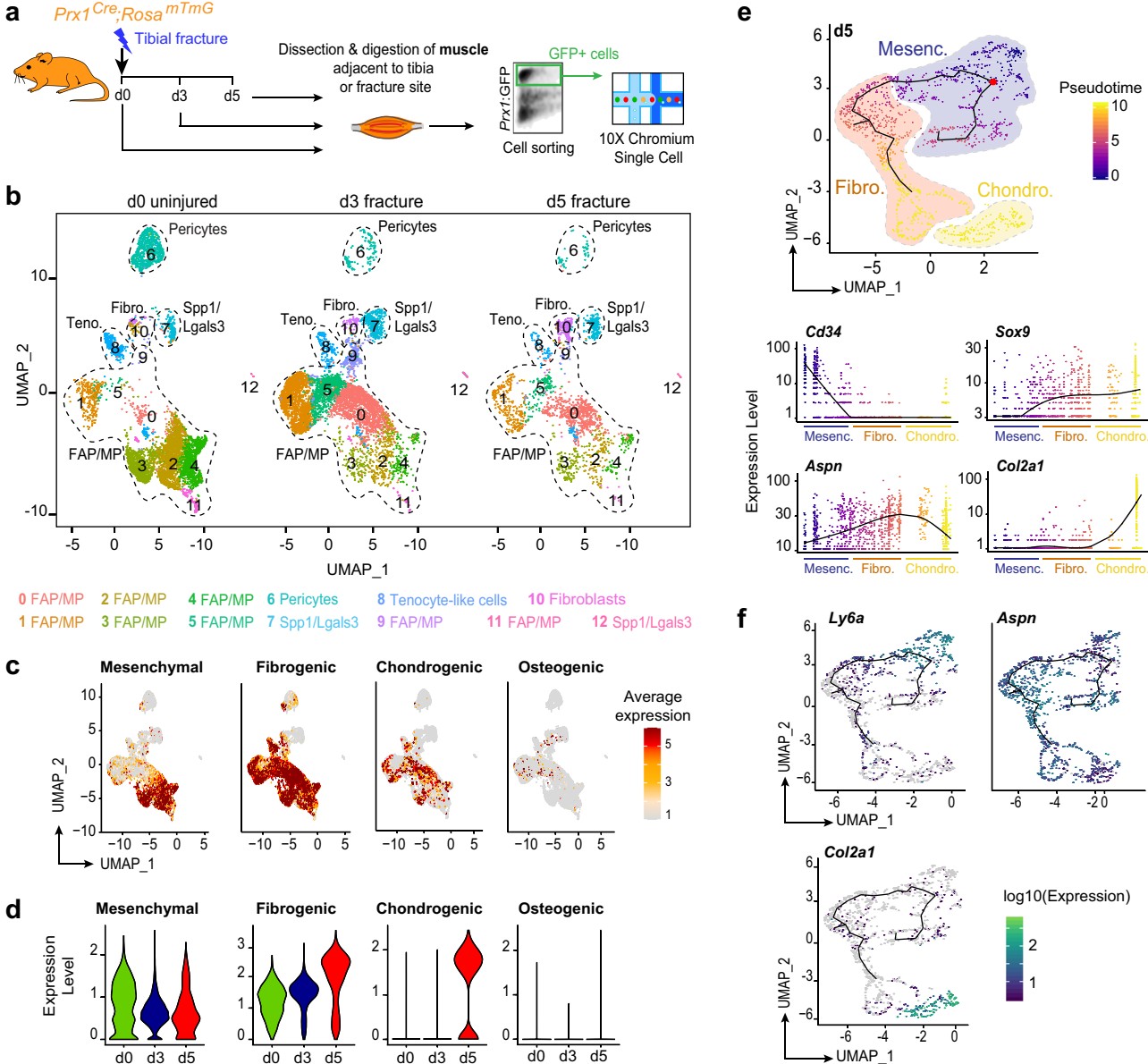

**Fig. 3 Single-cell analyses of skeletal muscle mesenchymal progenitors in response to bone fracture. a** Experimental design of scRNAseq analyses. Prx1-derived skeletal muscle cells were isolated from skeletal muscles surrounding the tibia at day 0, day 3, and day 5 post non-stabilized fracture and sorted based on GFP-expression prior to scRNAseq. **b** UMAP projection of color-coded clustering of integrated d0, d3, and d5 samples. Sub-populations are limited by a black dotted line and cluster identities are indicated below. Teno. tenocyte like cells, Fibro. fibroblasts. **c** FeaturePlot of mesenchymal, fibrogenic, chondrogenic and osteogenic lineages scores. **d** Expression of mesenchymal, fibrogenic, chondrogenic, and osteogenic lineages scores per sample. **e** Pseudotime analysis of FAP/MP at d5 post-fracture (top). Expression of *Cd34*, *Aspn*, *Sox9,* and *Col2a1* genes along pseudotime (bottom). Fibro. fibrogenic, Mesenc. mesenchymal, Chondro. chondrogenic. **f** FeaturePlot of *Ly6a*, *Aspn,* and *Col2a1* expression as markers of mesenchymal, fibrogenic, and chondrogenic lineages, respectively, in d5 post-fracture sample.

day 21 (Fig. 6c). To attenuate callus fibrosis, we treated mice with the clinically approved pan-tyrosine kinase inhibitor Imatinib® that among others inhibits receptor phosphorylation including PDGFR, BCR/ABL, and c-Kit[33,34]. Daily injection of Imatinib following polytrauma in a semi-stabilized fracture model reduced the volume of fibrous tissue and increased the volume of mineralized bone by d21 post-injury compared to mice treated with PBS. Imatinib treatment significantly increased bone bridging with 50% of fully healed calluses in the treated group compared to 10% in the PBS control group (Fig. 7). Imatinib treatment was also beneficial in the non-stabilized fracture model by accelerating cartilage and bone resorption and decreasing the volume of fibrous tissue at d21 compared to control (Supplementary Fig. 10).

Although we cannot exclude the impact of Imatinib treatment on other cell types beside Prx1-derived muscle resident cells, the results show that callus fibrosis originating for skeletal muscle can be decreased pharmacologically to improve bone repair after musculoskeletal trauma.

## Discussion

In this study, we uncover that skeletal stem/progenitor cells from multiple tissue origins cooperate to repair bone. These skeletal stem/progenitor cells reside not only in bone (bone marrow, periosteum) but also in adjacent skeletal muscle and are all derived from a common Prx1-derived mesenchymal lineage. Previous work supported that cellular contribution of

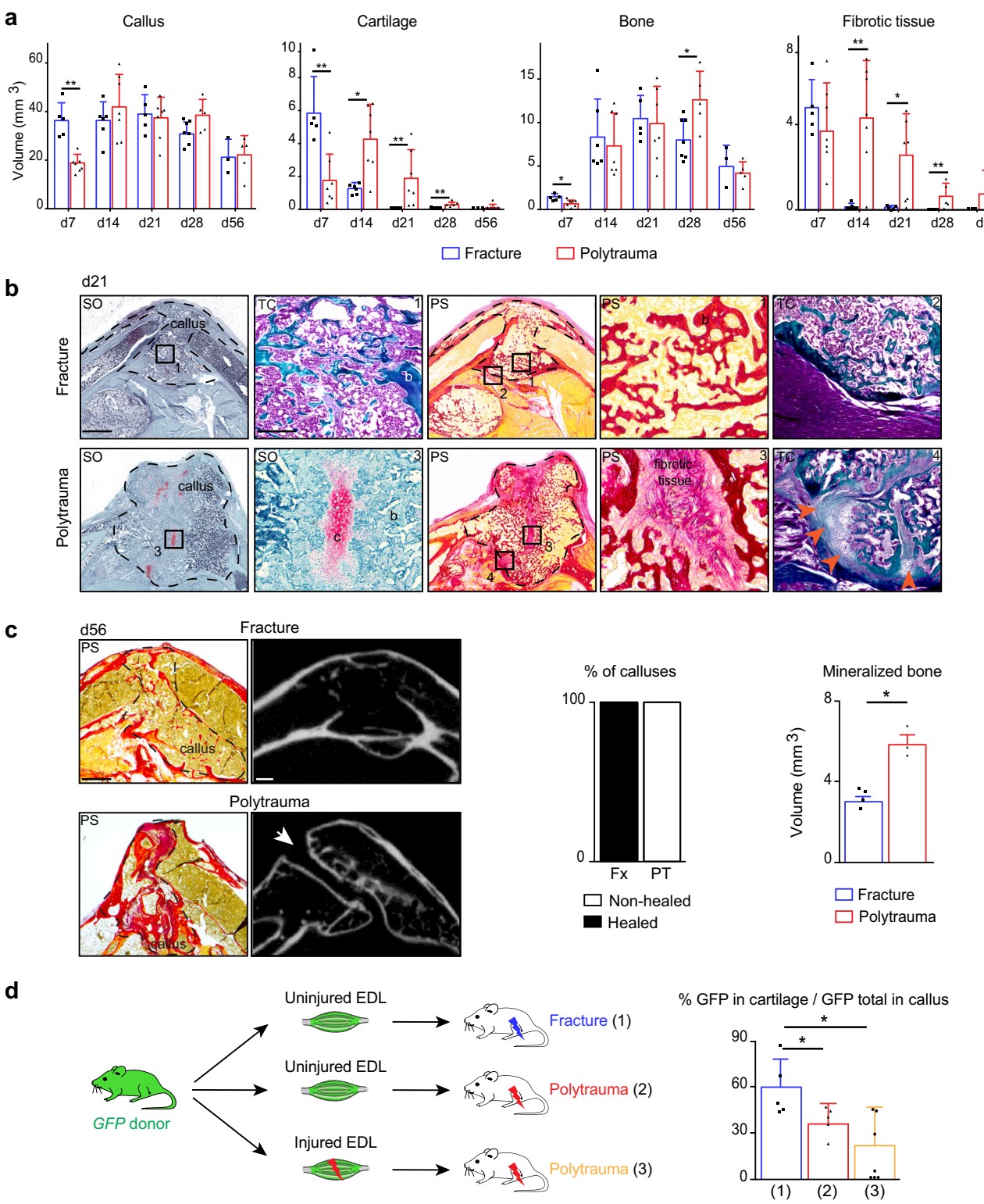

skeletal muscle to bone repair was observed in specific conditions, i.e. periosteal stripping, and that non-skeletal mesenchymal stromal cells lacked chondro-osteogenic properties[11,19,35]. Our results exclude the endogenous cellular contribution of the myogenic lineage during bone repair in vivo but demonstrate the direct contribution of Prx1-derived skeletal muscle MP to cartilage and bone during bone repair. These skeletal muscle MP localize within skeletal muscle interstitium and overlap with FAP/MP population. This reveals unknown functions of

FAP/MP as a heterogeneous and plastic population of cells that can adapt their fate according to the environment[3,19,25]. Using scRNAseq analyses, we show that within skeletal muscle MP, the FAP/MP population is distinct from pericytes, tenocyte-like and Spp1/Lgals3 populations and is the most responsive to bone injury. Skeletal muscle MP also express markers previously identified for skeletal stem/progenitor cells in bone[4,28–31]. Given the diversity and heterogeneity of MP, further analyses will be required to clarify the similarities and

**Fig. 4 Polytrauma impairs bone healing and cellular recruitment from skeletal muscle. a** Histomorphometric quantification of callus, cartilage, bone, and fibrotic tissue volumes after non-stabilized tibial fracture or polytrauma through all stages of bone repair (d7 $n = 5$ and $n = 7$; d14, $n = 6$ and $n = 7$, d21 $n = 5$ and $n = 7$, d28 $n = 7$ and $n = 5$, d56 $n = 3$ and $n = 5$ for fracture and for polytrauma respectively) (Exact $p$-value calculated with two-sided Mann–Whitney test: Callus d7: 0.0025; Cartilage d7: 0.0051, d14: 0.0189, d21: 0.0025, d28: 0.0013; Bone d7: 0.0101, d28: 0.0303; Fibrotic tissue d14: 0.0023, d21: 0.0303 d28: 0.0013). **b** Representative callus sections stained with Safranin'O (SO), Trichrome (TC), and PicroSirius (PS) at days 21 (b: bone) (callus delimited by a black dotted line). Fully ossified callus and bone bridging in fracture (boxes 1, 2). Unresorbed cartilage (c), fibrosis (box 3) and absence of bone bridging (box 4, orange arrowheads) in polytrauma. **c** Representative callus sections stained with PS and micro-CT images of calluses at d56 post-fracture and polytrauma. Absence of bone bridging at d56 is pointed by a white arrow. Quantification of healed and non-healed calluses at d56 post-fracture or polytrauma (fracture $n = 4$, polytrauma $n = 3$). Quantification of mineralized bone volume at d56 post-fracture or polytrauma (Fx: fracture $n = 4$, PT: polytrauma $n = 3$) (Exact $p$-value calculated with $t$ test with two-sided Welch correction: 0.0357). Scale bars: low magnification = 1 mm; boxed areas = 100 μm. **d** Experimental design of uninjured or injured EDL muscle grafts from *GFP-actin* mice transplanted at the fracture site of wild type hosts after fracture or polytrauma. Histomorphometric quantification of GFP+ signal in cartilage at day 14 post-injury ((1) $n = 5$, (2) $n = 5$, (3) $n = 7$) (Exact $p$-value calculated with two-sided Mann–Whitney test: (1) vs. (2): 0.0317, (1) vs. (3): 0.0303). All data represent mean ± SD. Images are representative of two independent experiments.

functionalities between mesenchymal progenitors located in skeletal muscle and bone.

After fracture, we show that skeletal muscle MP adopt a fibrogenic fate within 3 days post-injury associated with marked up-regulation of ECM genes. During this transient fibrogenic state, skeletal muscle MP express the master regulator of chondrogenesis *Sox9* but are not engaged into a chondrogenic fate until day 5 post-fracture. Interestingly, lineage tracing showed that skeletal muscle MP are recruited at the fracture site between day 5 and day 7, indicating that they are already committed to chondrogenesis within muscle tissue before migrating at the center of the callus. Skeletal muscle MP preferentially engage in the chondrogenic lineage in the early stages of callus formation and later give rise to osteoblasts through the endochondral ossification process. Endochondral ossification is required for bone repair highlighting the important role of skeletal muscle as a direct contributor to bone healing. Impairment of this cellular contribution from skeletal muscle may cause delayed bone healing in disease conditions or in trauma.

In a polytrauma environment, scRNAseq analyses reveal that activation of skeletal muscle MP into the transient fibrogenic state is impaired and cells fail to undergo chondrogenesis. At later stages of repair, damaged skeletal muscle adjacent to the bone fracture is responsible for fibrous tissue accumulation within the facture callus, interfering with fracture consolidation. This fibrotic phenotype caused by injured skeletal muscle is independent of the mechanical environment of the fracture as we observed similar results using non-stabilized and semi-stabilized fracture models. Skeletal muscle surrounding bone thus drives the fibrotic response and fibrotic remodeling during bone repair. Fibrosis is a dynamic process common to many tissue regeneration processes beginning with an initial phase of fibrotic response required for resident stem/progenitor cell activation and followed by active fibrotic remodeling necessary for completion of tissue regeneration[36–38]. In other regenerative processes, fibrotic progenitors are resident cell populations distinct from tissue-specific stem/progenitor cells[21,39–41]. In bone regeneration, we find that fibrotic progenitors are recruited from adjacent skeletal muscle after polytrauma and are derived from the same pool of progenitors that will undergo chondrogenesis. Several molecular therapies have been developed to treat fibrosis. Treatment using Imatinib®, a pan-inhibitor of PDGFR, BCR/ABL, and c-Kit signaling pathways, ameliorates the late stages of bone repair in our polytrauma model suggesting potential applications of Imatinib® in orthopedics. However, due to pleiotropic effects of Imatinib, other more specific kinase inhibitors and anti-fibrotic drugs may be further developed for therapeutic use. Overall, our findings have wide implications in musculoskeletal health, as they bring additional knowledge on the role muscle plays during bone repair

and define skeletal muscle MP as a prime target to enhance bone repair and prevent pathological fibrosis.

## Methods
Further information and requests for resources and reagents should be directed to and will be fulfilled by Colnot Céline (celine.colnot@inserm.fr).

**Mice**. C57BL/6ScNj, *beta-actinGFP* (GFP), *Prx1^Cre^*, *Pax7^CreERT2^*, *Rosa-tdTomato-EGFP* (Rosa^mTmG^) and *Rosa^YFP^* mice were obtained from Jackson Laboratory (Bar Harbor, ME) and maintained on a C57BL6/J background. Mice were bred and kept under controlled pathogen conditions in separated ventilated cages with controlled humidity an ambient temperature, with 12:12-hour light:dark cycles and free access to food and water in the animal facilities of IMRB, Creteil and Imagine Institute, Paris. All experiments were performed in compliance with procedures approved by the Paris Est Creteil and Paris University Ethical Committees. Animals used for all experiments were males or females 10–14-week-old and experimental groups were homogeneous in terms of animal gender and age. No specific randomization methods were used. Sample labeling allowed blind analyses.

**Tamoxifen injection**. To induce Cre recombination, Tamoxifen (TMX, T5648, Sigma) was prepared at a concentration of 10 mg/mL diluted in corn oil, heated at 60 °C for 2 h and injected intraperitoneally (3 injections, 300 μL per injection). *Pax7^CreERT2^*;Rosa^mTmG^ mice were injected with Tamoxifen at days 7, 6, and 5 before fracture.

**Tibial fractures and polytrauma**. In this study, two tibial fracture models were employed, (i) a non-stabilized fracture model in which the endochondral ossification process is predominant and better represents the contribution of skeletal muscle to bone healing, and (ii) a semi-stabilized fracture model, that also involves endochondral ossification and is more clinically relevant. For all surgical procedures, mice were anesthetized with an intraperitoneal injection of Ketamine (50 mg/mL) and Metedomidine (1 mg/kg) and received a subcutaneous injection of Buprenorphine (0.1 mg/kg) for analgesia. The right leg was shaved and cleaned using Vetidine soap and solution (VET 001, Vetoquinol). For non-stabilized fractures, the tibial surface was exposed, and the bone was cut to create the fracture via osteotomy. For semi-stabilized fractures, tibial fracture was first induced in the mid-diaphysis as described above. A pin was placed in the medullar cavity of the tibia from the knee to the ankle to stabilize the fracture site. For polytrauma, the skin was incised and skeletal muscles surrounding the tibia were separated from the bone. Mechanical injury to skeletal muscles, including *tibialis anterior* (TA), *tibialis posterior*, *extensor digitorum longus* (EDL), *soleus*, *plantaris*, *gastrocnemius* muscles surrounding the tibia, was applied by compressing the muscles for 5 s along their entire length using a hemostat in a standardized and reproducible procedure. Tibial non-stabilized or semi-stabilized fracture was then created in the mid-diaphysis by osteotomy as described above. For skeletal muscle injury only, the same procedure was applied without bone fracture. For partial skeletal muscle injury, the same procedure was performed by compressing the TA and EDL muscles only prior to fracture. At the end of the procedure, the skin was sutured using non-resorbable sutures (72-3318, Harvard Apparatus). Mice were revived with an intraperitoneal injection of atipamezole (1 mg/mL), kept on heated plate, and were allowed to ambulate freely. Mice received two supplementary doses of analgesia within 24 h post-surgery and were monitored daily.

**Imatinib treatment**. After polytrauma injury, mice received daily intraperitoneal injections of Imatinib® (50 mg/kg/day, Selleckchem, STI571) or vehicle (PBS) from the day of fracture until sacrifice.

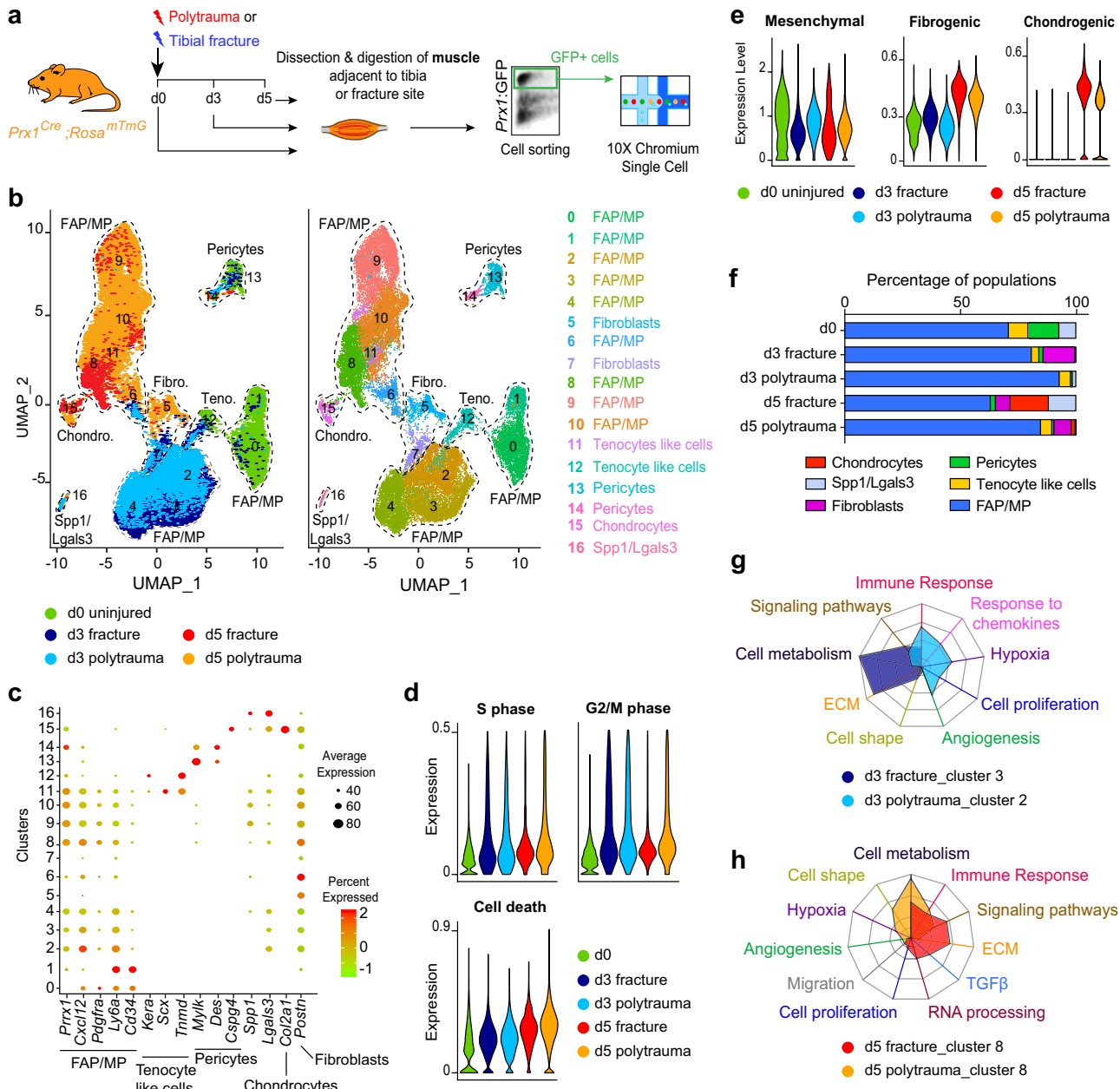

**Fig. 5 Single-cell analyses of skeletal muscle mesenchymal progenitors uncover impairment of initial fibrotic response in polytrauma. a** Experimental design of scRNAseq experiment. Prx1-derived skeletal muscle cells were isolated at d0, d3, and d5 post-non-stabilized fracture or post-polytrauma and sorted based on GFP-expression prior to scRNAseq. **b** UMAP projection of color-coded sample (left) and clustering (right) of integrated analysis of d0, d3, and d5 post-fracture and post-polytrauma samples. Sub-populations are delimited by a black dotted line. Sample and cluster identities are indicated below and on the right, respectively. **c** Dotplot of indicated gene expression identifying FAP/MP, tenocyte-like cell, pericyte, Spp1/Lgals3, chondrocyte, and fibroblast populations. **d** Expression of S and G2/M cell cycle phases and cell death scores per sample. **e** Pseudobulk expression of mesenchymal, fibrogenic and chondrogenic lineage score in d0, d3, and d5 post-fracture and post-polytrauma samples. **f** Percentage of each cell population per sample. **g** Radar chart of enriched Gene Ontology functions in cluster 3 corresponding to d3 post non-facture and cluster 2 corresponding to d3 post-polytrauma. **h** Radar chart of enriched Gene Ontology functions in d5 post-fracture versus d5 post-polytrauma in cluster 8. Teno. tenocyte like cells, Fibro. fibroblasts, Chondro. chondrocytes.

**EDL skeletal muscle transplantation**. Host mice received either a fracture or a polytrauma injury as described above. Donor mice were sacrificed by cervical dislocation and EDL muscle was dissected from tendon to tendon. The EDL muscle was transplanted adjacent to the fracture site at the time of fracture. EDL proximal tendon was sutured to the host patellar tendon and distal tendon was sutured to the host peroneus muscle tendon using non-resorbable sutures (FST, 12051-08). The skin was then sutured as described above. When fracture was induced one month after EDL muscle grafting, EDL muscle was first grafted as described above along the tibia without fracture. One month later, after skin incision, the tibial fracture was performed as described above via osteotomy after carefully separating the grafted EDL muscle and the bone.

**Isolation of skeletal muscle cells and cell transplantation**. *Prx1^Cre;Rosa^mTmG* mice were sacrificed by cervical dislocation. Skin and fascia were removed. TA, EDL, *tibialis posterior, plantaris, gastrocnemius*, and *soleus* muscles surrounding the tibia were dissected from tendon to tendon avoiding periosteum or bone marrow cell contamination. In a Petri dish with 1 mL of DMEM medium (21063029, Invitrogen), tendons and fat were removed, and skeletal muscles were minced using scissors. Skeletal muscles were transferred in 3 mL of digesting medium containing DMEM (21063029, Invitrogen), 1% Trypsin (210234, Roche), 1% collagenase D (11088866001, Roche) and incubated in a water bath at 37 °C for 2 h. Every 20 min individualized cells were removed and transferred into ice-cold growth medium containing αMEM (32561029, Life Technologies) with 1%

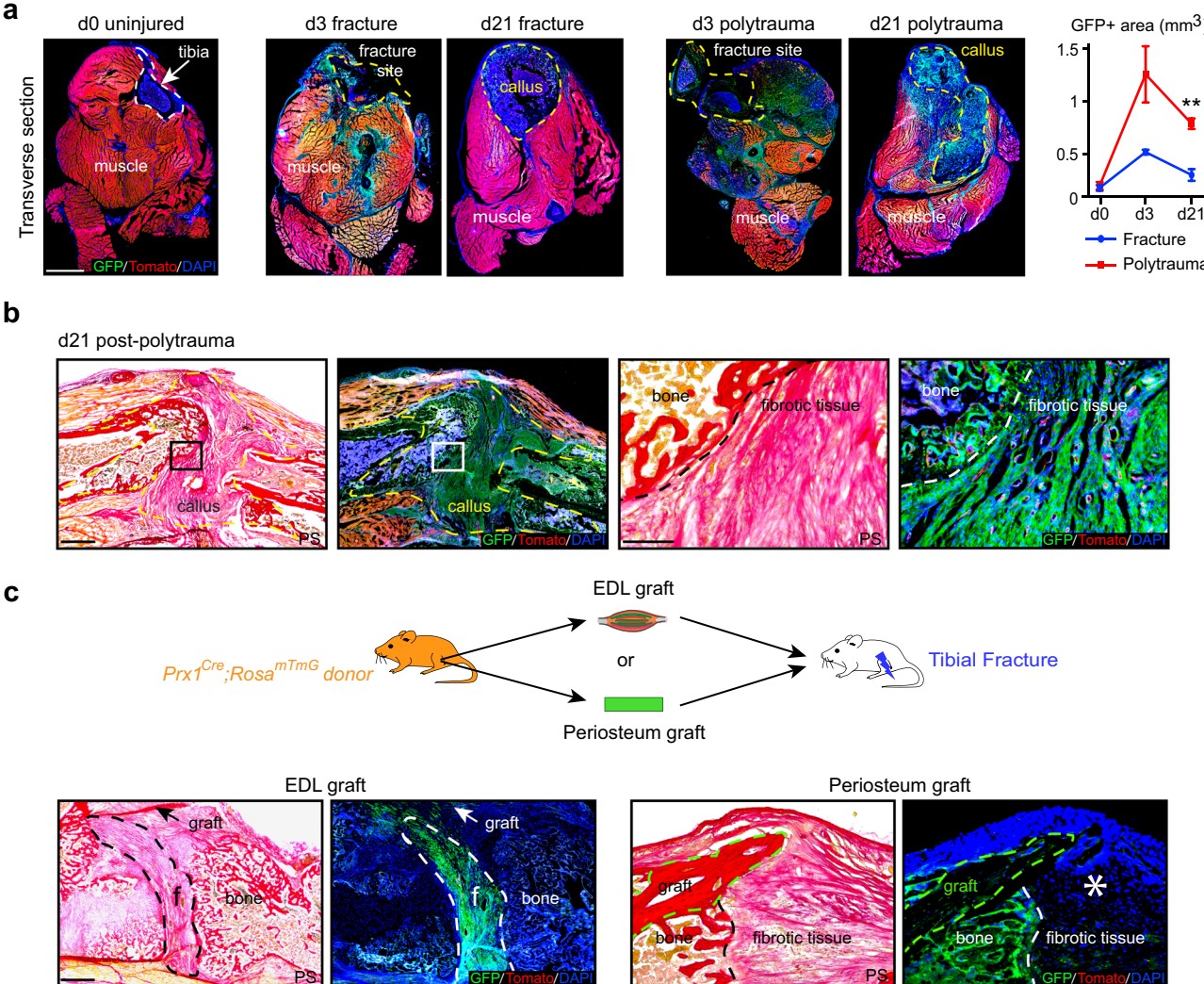

**Fig. 6 Callus fibrosis is produced by skeletal muscle mesenchymal progenitors in polytrauma. a** Left, Transverse sections of hindlimb from *Prx1^Cre; Rosa^mTmG* mice at day 0 (tibia bone delimited by a white dotted line), day 3 or d21 post-non-stabilized fracture or post-polytrauma (fracture site at d3 post-injury and callus at d21 post-injury delimited by yellow dotted line). Representative images of three distinct samples. Right, Quantification of GFP+ area within skeletal muscle (callus excluded) on transverse sections of hindlimb of *Prx1^Cre;Rosa^mTmG* mice at d0 (uninjured), d3 and d21 post-fracture and post-polytrauma (d0 n = 4, d3 post-fracture n = 3, d3 post-polytrauma n = 3, d21 post-fracture n = 4 and d21 post-polytrauma n = 3). Exact p-value calculated with t test with two-sided Welch correction: 0,0013. Scale bar: 0.5 mm. **b** Left. Longitudinal sections of fracture callus at day 21 post-polytrauma in *Prx1^Cre; Rosa^mTmG* mice stained with Picrosirius (PS) and adjacent sections counterstained with DAPI (callus delimited by a dotted line). Right, High magnification of boxed areas (dotted lines mark the limit between bone and fibrotic tissue). **c** Top: Experimental design of EDL muscle or periosteum grafts from *Prx1^Cre; Rosa^mTmG* mice transplanted at the fracture site of wild type hosts after polytrauma. Bottom: Longitudinal callus sections stained with PS and adjacent sections counterstained with DAPI at day 21 showing presence of GFP+ cells derived from EDL muscle graft in fibrotic tissue (delimited with a white dotted line) and presence of GFP+ cells derived from periosteum graft (delimited with a green dotted line) in bone but absence in fibrotic tissue (asterisk). **a–c** Representative images of three independent experiments. DAPI in blue, GFP in green, Tomato in red. f fibrotic tissue. **b–c** scale bars: low magnification:1 mm, high magnification: 300 μm. All data represent mean ± SD.

penicillin–streptomycin (P/S, 15140122, Life Technologies), 20% lot-selected non-heat-inactivated fetal bovine serum (FBS, 10270106, Life Technologies) and 10 ng/ml bFGF (3139-FB-025/CF, R&D) and fresh digesting medium was added to the undigested tissue. This step was repeated until all skeletal muscle was digested. Cells were then filtered through 100 μm filters (352360, Dutscher) and 40 μm filters (352340, Dutscher), centrifuged 10 min at 300×g and resuspended in 3 mL of growth medium.

For cell sorting, skeletal muscle cells were resuspended in 1 mL of sorting medium containing DMEM medium (21063029, Invitrogen) with 2% of FBS and 1% of P/S. Cell viability marker Sytox blue (1/1000, S34857, Thermofischer) was added just before sorting. Cell sorting was performed on BD FACS Aria II SORP (BD Biosciences) and cells were collected in growth medium. For cell transplantation, 150,000 freshly sorted skeletal muscle cells were embedded in TissuCol® kit TISSEEL (human fibrogen 15 mg/mL and thrombin 9 mg/mL, Baxter, France) according to manufacturer's instructions[5]. Open fracture was performed as

described above and cell pellets were implanted at the fracture site at the time of fracture.

**Micro-computed tomography analysis**. Samples were imaged using X-ray micro-CT device (Quantum FXCaliper, Life Sciences, Perkin Elmer, Waltham, MA). The X-ray source was set at 90 kV and 160 μA. Tridimensional images were acquired with a field of view of 10 mm and an isotropic voxel size of 20 μm. Full 3D high-resolution raw data were obtained by rotating both the X-ray source and the flat panel detector 360 around the sample using DataViewer64 software v1.5.2.4 (Bruker microCT, Hamburg, Germany). The volume of mineralized bone in callus was quantified by delimitating the callus area excluding the cortices using CTan software v1.17.7.2 (Bruker, Hamburg, Germany). Callus bridging was determined as the number of bridged cortices on the dorsal and ventral sides, in the sagittal and longitudinal axes. A fracture was considered to be fully healed when cortices

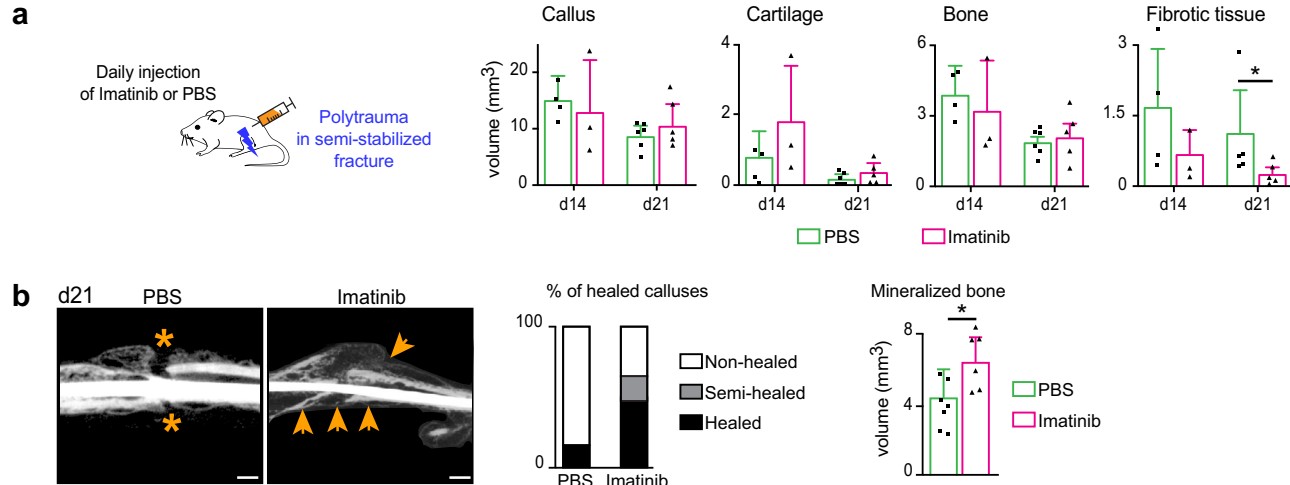

**Fig. 7 Imatinib decreases callus fibrosis and improves fracture healing after polytrauma. a** Daily injection of Imatinib® (50 mg/kg/day) or vehicle (PBS) after polytrauma in semi-stabilized tibial fracture. Histomorphometric analyses of total callus, cartilage, bone, and fibrotic tissue volumes of PBS-treated and Imatinib-treated mice at days 14 and 21 (d14 PBS-treated $n = 4$, d14 Imatinib-treated $n = 3$, d21 PBS-treated $n = 6$, d21 Imatinib-treated $n = 5$) (Exact $p$-value calculated with two-sided Mann–Whitney test: Fibrotic tissue d21: 0.0317). **b** Representative micro-CT images of fracture calluses in PBS-treated and Imatinib-treated mice. Absence of bone bridging in PBS-treated mice (orange asterix) and presence of bone bridging in Imatinib-treated mice (orange arrow). Quantification of healed and non-healed calluses at d21 post-treatment (PBS-treated $n = 7$, Imatinib-treated $n = 6$). Quantification of mineralized bone volume (right) (PBS-treated $n = 7$, Imatinib-treated $n = 6$) (Exact $p$-value using two-sided Mann–Whitney test: Mineralized bone d21: 0.0303). **b**: Representative images of three independent experiments. scale bars: 1 mm. All data represent mean ± SD.

bridged at 3 or 4 sites; semi-healed when cortices bridged at 2 sites and non-healed when cortices bridged at 1 or 0 site.

**Callus sample processing, histology, and histomorphometric analyses**. Mice were sacrificed by cervical dislocation and fractured tibias were harvested at days 3, 5, 7, 14, 21, 28, or 56 post-surgery according to the experiment. Samples were fixed 24 h in 4% PFA (15714, Euromedex) and decalcified in 19% EDTA (EU00084, Euromedex) for 21 days at 4 °C upon agitation. Samples with endogenous expression of fluorescent reporter proteins were incubated in sucrose 30% at 4 °C over night and then embedded in OCT (F/62550-1, MMFrance). All other samples were embedded in paraffin. All samples were sectioned throughout the entire callus and consecutive sections were collected. Tissue sections from paraffin-embedded samples were incubated in NeoClear® (1098435000, VWR) for 2 × 5 min and rehydrated in successive baths of 100%, 90%, and 70% ethanol and then rinsed in PBS for 5 min. Frozen sections were dried at room temperature for 30 min and rehydrated in PBS for 10 min. After staining, sections were dehydrated in 70%, 90%, 100% ethanol for 3 min each and in NeoClear® for 10 min. Slides were mounted using NeoMount® mounting medium (1090160100, VWR).

*Safranin'O staining*. Nucleus were stained with Weigert's solution of 5 min. Slides were next rinsed with tap running water for 3 min and then stained into 0.02% Fast Green for 30 sec (F7252, Sigma), followed by 30 sec into 1% acetic acid. To detect proteoglycan within cartilage, slides were stained with safranin'O solution for 45 min (S2255, Sigma).

*Masson's trichrome staining*. Sections were stained with Harris haematoxylin (dilution ½) for 5 min (F/C0283, MMFrance), rinsed in running tap water 5 min, then dyed with Mallory red for 10 min, rinsed for 5 min and then differentiated into phosphomolybdic acid 1% for 10 min (HT153, Sigma). Collagen fibers were stained with light green for 20 min (720-0335, VWR) and fixed into acetic acid 1%.

*Picrosirius staining*. Sections were stained in PicroSirus solution (0.1 g of Direct Red 80, 43665-25G, Sigma diluted into 100 mL of saturated solution of picric acid, 80456, Sigma) for 2 h at room temperature, protected from light.

For histomorphometric analyses, every 30th section throughout the entire callus was stained with Safranin'o, Masson's Trichrome, Picrosirius or counterstained with DAPI to visualize GFP signal and pictured using a Zeiss Imager D1 AX10 light microscope (Carl Zeiss Microscopy GmbH)[5]. Areas of callus, cartilage, bone, fibrosis, and GFP signal were determined using ZEN software v1.1.2.0 (Carl Zeiss Microscopy GmbH) and volumes were calculated via the following formula: Volume $= \frac{1}{3} h \sum_1^{n-1}(Ai + A(i+1) + \sqrt{Ai * A(i+1)})$ where $Ai$ and $Ai+1$ were the areas of callus, cartilage, bone, fibrosis or GFP signal in sequential sections, $h$ was the distance between $Ai$ and $Ai+1$ and equal to 300 μm, $n$ was the total number of sections analyzed in the sample.

**Skeletal muscle sample processing and analyses**. TA skeletal muscle samples were harvested at specific time points, fixed for 3 h in PFA 4%, incubated in sucrose 30% for 2 h and embedded in OCT for cryosection. Sections were collected in the center of the muscle.

*Haematoxylin–Eosin staining*. Slides were stained with hemalun for 3 min, rinsed in running tap water for 1 min, stained with eosin for 1 min (6766009, Thermo Fischer Scientific), then rinsed in water and pictured using a Zeiss Imager D1 AX10 light microscope (Carl Zeiss Microscopy GmbH).

**GFP quantification**. Tibial samples from *Prx1^{Cre}*;Rosa^{mTmG} mice were processed as described above for cryosection and were transversally embedded in OCT. Every fifth transverse section was collected in the middle of the diaphysis of uninjured tibia, at the fracture site day 3 post-fracture or in the center of the callus at day 21 post-fracture. Sections were counterstained with DAPI (eBiosciences, 495952). Entire transverse sections were pictured using Spinning Disk (Carl Zeiss Microscopy GmbH) and GFP signal was quantified within skeletal muscle excluding callus area using ZEN software v1.1.2.0 (Carl Zeiss Microscopy GmbH).

For GFP signal quantification of GFP-EDL muscle graft contribution, area of fibrous callus, endochondral bone and intramembranous bone were defined based on cell morphology and GFP+ cells were manually counted in each area every 10th slide through the entire callus. Number of GFP+ cells in each area was then normalized over the total number of GFP+ cells.

**Immunofluorescence**. GFP and Tomato signals were detected without immuno-fluorescence staining. Cryosections were dried at room temperature for 30 min, rehydrated in PBS for 10 min and then mounted with Fluoromount (eBiosciences, 495952)[5].

For calluses samples, paraffin-embedded sections were deparaffinized and rehydrated as described above. For Periostin immunofluorescence, sections were blocked in 5% donkey serum for 1 h and incubated in goat anti-periostin antibody (1/400, ref AF2955 R&D) diluted in 5% donkey serum in PBS (D9663, Sigma) over-night at 4 °C. Sections were then washed in PBS 3 × 5min, incubated with donkey anti-goat AF488 antibody (1/250, A11055, Invitrogen) diluted in 5% donkey serum then rinsed with PBS for 10 min and mounted using Fluoromount. For PDGFRα immunofluorescence, antigen retrieval was performed using sodium citrate buffer at 95 °C for 20 min. Slides were then cooled down in sodium citrate buffer on ice for 20 min, rinsed in PBS 2 × 10min and then incubated in blocking solution containing 5% normal goat serum (NGS, Ab7481, Abcam), 0.5% Triton (T8787, Sigma) in PBS for 1 h. Sections were washed in PBS 3 × 5min and incubated with goat anti-PDGFRα antibody (1/200, ref AF1062 R&D) diluted in blocking solution over-night at 4 °C. Sections were then washed in PBS 3 × 5 min, incubated with donkey anti-goat AF488 antibody (1/250, A11055, Invitrogen) diluted in blocking buffer then rinsed with PBS for 10 min and mounted using Fluoromount.

For COLX, OSX, and S100b immunofluorescence, cryosections were dried at room temperature for 30 min and then rehydrated 15 min in PBS. Sections were then blocked for 30 min in blocking buffer PBS-Triton 0.25–5% NGS and incubated with rabbit anti-COLX (1/200, ref ab58632 Abcam), rabbit anti-OSX (1/400, ref ab22552 Abcam), or rabbit anti-S100b (1/200, ref Z0311, Agilent) for 45 min at room temperature. Slides were then washed in PBS 3 × 5 min and incubated with goat anti-rabbit AF647 (1/250, A21245, LifeTechnologies) or goat-anti-rabbit AF546 (1/250, A11010, Invitrogen), respectively, diluted in blocking buffer for 45 min at room temperature. Slides were then washed in PBS 3 × 5 min and mounted using Fluoromount. Samples were pictured using a Zeiss Imager D1 AX10 light microscope (Carl Zeiss Microscopy GmbH).

For skeletal muscle samples, cryosections were dried for 30 min at room temperature protected from light and then rehydrated for 10 min in PBS. For anti-NG2, anti-PDGFRα, and anti-PDGFRβ immunofluorescence, cryosections were post-fixed in PFA 4% for 5 min, rinsed 3 × 5 min in 0.5% Triton in PBS, blocked in 5% NGS and 0.5% Triton diluted in PBS and incubated over night at 4 °C with primary antibody: rabbit anti-NG2 (1/50, AB5320, Merck), goat anti-PDGFRα (1/200, AF1062, R&D), or rabbit anti- PDGFRβ (1/200, ab32570, Abcam). Slides were rinsed in PBS 3 × 5 min and then incubated for 1 h at room temperature in goat anti-rabbit AF647 (1/250, 21245, Life Technologies) or donkey anti-goat AF647 (1/500, ref ab150135 Abcam). Slides were mounted with Fluoromount (ref 495952, eBiosciences). For anti-CD29 immunofluorescence, cryosections were post-fixed in PFA 4% for 10 min, washed for 2 × 5 min in PBS, permeabilized in PBS-Triton 0.25%, blocked in 1% BSA for 15 min (A2153, Sigma) and incubated with goat anti-mouse CD29 (1/50, 026202, R&D) overnight at 4 °C. Slides were next rinsed and incubated in donkey anti-goat AF647 (1/500, ab150135, Abcam). Slides were mounted with Fluoromount. All pictures were obtained using a LSM700 confocal microscope (Carl Zeiss Microscopy GmbH) and processed with ImageJ v2.1.0.

For anti-αSMA immunocytofluorescence, cells were washed with PBS and fixed in PFA 4% for 15 min, rinsed in PBS, permeabilized in PBS–Triton 0.25%, blocked in 5% NGS, incubated with anti-αSMA-Cy5 (1/400, ref AC12-0159-11, clone 1A4, Clinisciences) for 1 h and mounted with Fluoromount. Pictures were taken using a Zeiss Axio Vertical A1 light microscope (Carl Zeiss Microscopy GmbH).

**Flow cytometry**. For flow cytometry analysis, skeletal muscle cells were incubated with CD31-PECy7 (1/400, PECAM-1, 561410, BD Biosciences), CD45-PECy7 (1/400, leukocyte common antigen, Ly-5, 552848, BD Biosciences), CD11b-PECy7 (1/400, integrin αM chain, 552850, BD Biosciences), CD34-AF700 (1/400, 560518 BD Biosciences), Sca1-BV650 (1/400, 740450, BD Biosciences), CD29-APC (1/400, 130-102-557, Miltenyi Biotec), PDGFRα-BV711 (1/200, 740740, BD Biosciences), and PDGFRβ-APC (1/200, 17-1402-80, Invitrogen) for 15 min on ice protected from light. Cells were then washed by adding 1 mL of sorting medium and centrifuged for 10 min at 300×g. Supernatant was trashed and cell pellets were resuspended in 200 μL of sorting medium and Sytox blue (1/1000, S34857, Thermofisher) was added just before analysis. Compensation beads (01-2222-42, Thermo Fischer) were used for initial compensation set up and FMO controls were used to determine background level of each color. Analyses were performed on BD LSR Fortessa X-20 using BD FACS Diva Software v8 (BD Biosciences) and results analyzed using FlowJo, LLC software, v10.5.3. Gating strategy used for analyses is available in Supplementary Fig. 4b.

**In vitro differentiation**. After cell sorting, Prx1-derived skeletal muscle cells were expanded in six-well plates to allow in vitro differentiation.

For adipogenic differentiation, sub-confluent cells were placed into adipogenic medium containing αMEM supplemented with 10% FBS, 0.1 μg/ml insulin (I3536, Sigma), 100 μM indomethacin (I7378, Sigma), 0.5 mM 3-isobutyl-1-methylxantine (I5879, Sigma), and 0.1 μM dexamethasone (D8893, Sigma). Medium was changed every 3 days for 2 weeks. Lipid droplets were stained with Oil Red O solution (O0625, Sigma) and nuclei with Harris haematoxylin solution (F/C0283, MMFrance). For osteogenic differentiation, cells at confluence were cultured in osteogenic medium containing αMEM supplemented with 10% FBS, 0.1 μM dexamethasone, 0.2mM L-ascorbic acid (A8960, Sigma), and 10 mM glycerol 2-phosphate disodium salt hydrate (G9422, Sigma). Medium was changed every 3 days for 3 weeks. Mineralization was revealed with 0.2% alizarin red staining (A5533, Sigma). For chondrogenic differentiation, cells were plated as micromass at a concentration of $5 \times 10^5$ cells in 200 μL of growth media. 2 h later, growth medium was replaced by chondrogenic medium corresponding to DMEM with 10% FBS with 0.1 μM dexamethasone, 100 μg/mL sodium pyruvate (P5280, Sigma), 40 μg/mL L-proline (P0380, Sigma), 50 μg/mL L-ascorbic acid, 50 mg/mL Insulin-Serine-Transferase (I1884, Sigma), and 10 ng/mL TGFβ1 (T7039, Sigma). After 3 days, cartilage matrix was stained with alcian blue (A5268, Sigma). For myogenic differentiation, Prx1-derived muscle cells were plated at 1000 cells per cm² and induced with myogenic medium containing F10 (31550-02, Life Technologies), 2% horse serum (26050088, Life Technologies) and 1% P/S for 3 days. For fibrogenic differentiation, cells were grown until sub-confluence and induced to fibrogenic differentiation with DMEM high-glucose (10566016, Life Technologies) with 10% FBS, 1% P/S, and TGF-β1 at 1 ng/mL. All pictures were obtained with a Leica DM IRB light microscope.

**RTqPCR analyses**. Prx1-derived skeletal muscle cells at 80% of confluence were dissociated using trypsin, pelleted for 10 min at 300×g and frozen at −80 °C. RNA extraction was performed with RNAeasy Kit (74134, Qiagen) following manufacture's instructions. Amount of RNA was quantified using NanoDrop 2000 UV–Vis spectrophotometer (Thermo Scientific). 500 μg of RNA were used to synthetize cDNA. RNAs were mixed with 1 μL of oligo$_{12–18}$ (18418-012, Life Technologies) and 1 μL 10 mM dNTP Mix (18427-013, Life Technologies) and heated at 65 °C for 5 min and left on ice for 1 min. Next, 4 μL 5× first-strand buffer, 1 μL 0.1 M DTT, 1 μL Superscript III RT® (18080-044, Life Technologies), and 1 μL RNaseOUT® (10777-019, Life Technologies) were added and incubated at 50 °C for 1 h. The reaction was inactivated by heating at 70 °C for 15 min. qPCR mix was composed by 1 μL of primers, 4 μL of RNAse free H$_2$O, 10 μL of SYBR green Master Mix (11744-100, Life Technologies), and 5 μL of cDNA. qPCR reaction was performed using 7300 Real-Time PCR System v.1.3.1 (Thermofischer Scientific). Mouse *Gapdh* was used as internal calibrator. qPCR analyses were done following ΔΔCT methods on Excel v14.7.3. Primers sequences are provided in the supplementary information, Table 3.

**Single-cell RNAseq analyses**. Prx1-derived skeletal muscle cells were isolated as described above from *Prx1$^{Cre}$;Rosa$^{mTmG}$* d0 mice (un-injured), at day 3 post non-stabilized fracture, day 3 post-polytrauma, day 5 post non-stabilized fracture or day 5 post-polytrauma. Two mice were used per sample and only skeletal muscle was dissected. Periosteum and bone marrow were not collected. The scRNAseq libraries were generated using Chromium Single Cell 3'Library & Gel Bead Kit v.2 (10× Genomics) according to the manufacturer's protocol. Briefly, cells were counted, diluted at 1000 cells/μL in PBS + 0.04% FBS and 20,000 cells were loaded in the 10× Chromium Controller to generate single-cell gel-beads in emulsion. After reverse transcription, gel-beads in emulsion were disrupted. Barcoded complementary DNA was isolated and amplified by PCR. Following fragmentation, end repair and A-tailing, sample indexes were added during index PCR. The purified libraries were sequenced on NovaSeq 600 (Illumina) with 26 cycles of read 1, 8 cycles of i7 index and 98 cycles of read 2.

*Data preprocessing*. Fastq files from the scRNA 10X libraries were processed using the CellRanger Count pipeline with its default parameters (v5.0.1). Reads were aligned against the mm10 reference genome. The RNA data quality control and downstream analysis were performed using the Seurat R package (v3.0.2) and the standard Seurat v3 integration workflow.

*Aggregate sample generation*. We generated aggregate sample to compare d0, d3 post-fracture, d5 post-fracture, d3 post-polytrauma, and d5 post-polytrauma in a common dataset. Aggregate sample was generated according to cell ranger aggr pipeline in order to remove batch effect due to sequencing depth.

*Seurat analysis*. Seurat v3.1.2 and Rstudio v1.2.1335 were used for analysis of scRNA-seq data[42,43]. Cells expressing between 350 and 8000 genes and expressing <20% of mitochondrial gene were retained for analysis, genes expressed in <5 cells were not taken into account. Clustering was performed using the first 20 principal components with 0.5 as resolution and clusters were visualized using UMAP projection. Integrated analysis of d0, d3, and d5 post-fracture was performed using top 2000 features and the 20 first principal components with a resolution set at 0.5. Differentially expressed genes were determined using Wilcoxon rank sum test with *P*-value < 0.05. For Gene Ontology (GO) analyses, differentially expressed genes were used to find enriched functions using Enrich R software (https://amp.pharm.mssm.edu/Enrichr/)[44,45]. GO functions including <5 genes and with adjusted *P*-value > 0.05 were excluded. GO functions were classified manually into global functions and the percentage of each function across all the functions found was plotted into radar graph as represented in the figure.

*Monocle analysis*. Monocle3 v0.2.3.0 was used for pseudotime analysis of FAP/MP of d5 post-fracture sample. Cells were ordered in a semi-supervised manner on the basis of Seurat clustering. Starting point corresponds to the highest expression of *Ly6a* and *Cd34* genes.

*Cell cycle analysis*. Cell cycle analysis was performed using Cell Cycle Regression vignette from Seurat package.

*Lineage analysis*. Signature score was calculated for each cell as arithmetic mean of the expression of the associated genes in each cell (Supplementary information, Table 1) and implemented as metadata in Seurat object.

*Integration with available data*. Our dataset was integrated with databases from Giordani et al. [25], Wilder-Scott et al. [23] and Tabula Muris[26] using the Seurat v3 package and standard workflow with 20 dimensions and 0.5 as resolution.

**Statistical analyses**. Data are presented as mean ± s.d. and were obtained from at least two independent experiments and *n* represents the number of samples used for the analysis. Statistical significance was determined with two-sided

Mann–Whitney test or *t*-test with Welch correction and reported from GraphPad Prism v6.0a. Differences were considered to be significant when $P < 0.05$.

**Reporting summary**. Further information on research design is available in the Nature Research Reporting Summary linked to this article.

## Data availability

Data that support findings of this study have been deposited in the Gene Expression Omnibus (GEO) database under the accession number GSE164573. Data from accession number GSE110878, GSE110038, and form Tabula Muris Consortium (https://tabula-muris.ds.czbiohub.org/) were used in Supplementary Fig. 5. Source data are provided with this paper.

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

## Acknowledgements

We thank M. Garfa-Traoré, N. Goudin, C. Cordier, O. Duchamp de Lageneste, S. Alonso-Martin, B. Drayton, C. Masson, S. Perrin, J. Collis, C. Alms, R. Prota for advice and/or technical assistance. In vivo imaging was performed at the Life Imaging Facility of Paris University (Plateforme Imageries du Vivant "Micro-CT platform" L. Slimani and F. Tilotta). Single-cell RNA sequencing was performed at the Genomics and Bioinformatics platform of Imagine Institute, INSERM UMR 1163, Paris. This work was supported by INSERM ATIP-Avenir to C.C. and M.M., Fondation de l'Avenir to C.C., ANR-13-BSV1-001 to C.C. and F.R., NIAMS R01 AR057344 and R01 AR072707 to C.C. and T. Miclau. A.J. was supported by a Ph.D. fellowship from Paris University. Micro-CT platform was supported by France Life Imaging (grant ANR-11-INBS-0006) and Infrastructures Biologie-Santé.

## Author contributions

A.J. performed the experiments with the help of A.K. and E.M.-S. J.M. assisted with flow cytometry analyses. M.L. performed scRNAseq libraries. M.M. and F.R. reviewed the manuscript and gave advice. A.J. and C.C. designed the experiments, analyzed the data, and wrote the manuscript. C.C. conceived the project and supervised the study.

## Competing interests

The authors declare no competing interests.
