## [Peer Review File · Nature Communications]

Reviewers' comments:

Reviewer #1 (Remarks to the Author):

In this manuscript, Anais et al. evaluate the contribution of muscle-resident Prx1-lineage cells to fracture repair. Overall the topic of which cells mediate fracture repair is of interest, especially given our increasingly granular understanding of skeletal populations. This manuscript has several methodological and conceptual strengths that are worth noting. First, consideration of combined bone/muscle injury and the mechanisms underlying why outcomes after this may diverge from bone injury only is a novel concept within the mechanistic bone biology literature. It is likely that this concept and the accompanying model developed here will be of broader relevance and interest for the field. There are also some methodologic strengths to note. First, there is an attempt to comprehensively characterize the muscle resident PRX1-lineage cells, though there are a number of specific concerns with elements of this approach as described below. Second, the use of multiple experimental timepoints for scRNA-seq studies strengthens the pseudotime analysis performed. Lastly, a transplantation system is used that provides strong evidence that the PRX1-lineage cells tracked are truly an extraskeletal resident population. Additionally, the identification of a possible source for fibrotic tissue present in pseudoarthroses/non-unions is of interest and has potential clinical relevance. Despite these strengths, a number of important weaknesses are noted that will be critical to address. Chief among these is 1) that the evidence for the contribution of the muscle resident Prx1-lineage cells to fracture healing is not currently convincing. Second, the outcome measures from fracture models utilized are mostly histology based, whereas uCT or other methods provide more robust and convincing data. All of these issues are described in detail below.

Major points:

1. The most important overall limitation in the manuscript is that the evidence that Prx1-lineage muscle resident cells have an important contribution to fracture healing is currently not convincing. The key experiment to address this central point of the manuscript is in Fig 1c (and extended fig 1c). This shows that the GFP+ Prx1-lineage cells appear to be largely restricted to peripheral portions of the callus region at days 7 and 14. From the images provided, it is unclear if the Fig 1c GFP+ cells are truly in the fracture callus or largely restricted to peripheral connective tissue. Additionally, none of the GFP+ cells in the day 14 samples (Fig 1c) show convincing chondrocyte morphology, and no co-labeling with cartilage markers is provided. No quantitative measure of the relative amounts of each of these cellular compartments that are reconstituted with Prx1-lineage cells is provided. The pictures presented give the impression that much of this engraftment is transient given that what appears to be a single cell is all that is shown in the day 28 image. This single cell at day 28 is not convincing for contribution to osteoblast lineage cells. Overall, the use of a great number of high power inserts throughout Fig 1 without reference to a lower power image (with the exception of Fig 1b) lessens the impact of this data. More direct clarification of whether a contribution to osteogenic lineages is claimed would be helpful, as well as a comparative discussion of the results of Fig 1a vs Fig 1b vs Fig 1c. From the model presented, one would expect fig 1c to give the same results as Figs 1a and b, but the data shown gives a different impression, especially as regards the contribution to chondrogenesis. Importantly, there is no direct functional evidence for the contribution of Prx1-positive cells to fracture healing, and claims about the role of muscle resident Prx1-positive cells are much weaker without supporting functional data.

2. A major limitation of the fracture studies is that primarily histologic endpoints are used. While histology is essential, especially for evaluating fibrous and cartilaginous callus tissue, uCT is still needed, as this is the most technically robust method to assess for overall delays in bone formation and for lack of bone bridging/non-union phenotypes. This issue is especially important as the non-stabilized fracture model utilized here has a very high degree of variability. Thus, robust quantitative measures (such as absolute measures of callus mineralization or uCT determined bridging/non-union

rates) needed to ensure that variability in the model is accounted for. Histomorphometry does not provide as convincing data due to the sampling issues inherent in histologic approaches. uCT does not need to be universally applied, but key results should be confirmed by targeted uCT analysis, such as they key groups used to argue for altered overall skeletal healing in the polytrauma and imatinib models.

3. A control, or at least reference to literature examining this point, showing that Prx1 cre is not deleting in myogenic lineages in the initial donors is needed.

4. The choice of the unstabilized fracture model is unusual, as stabilized models are usually selected due to both the lesser degree of variation present and, arguably, due to stabilized fractures being a more overall clinically relevant model. Commentary on the rationale for selecting an unstabilized model would be helpful. Given the use of an unstabilized model, there is a concern that the results of the polytrauma model could be entirely due to impairing the ability of surrounding muscle to stabilize the fracture, and not due to any specific impact of the muscle trauma on muscle resident mesenchymal progenitors. Changes in the degree of fracture site stabilization can have very large impacts on outcomes in fracture models, as often impacts from changes in the degree of stabilization that are bigger than the impact of biologic factors under study.

Minor points

1. The term EDL (extensor digitorum longus?) should be defined upon first use.

2. FACS plots should not be displayed as histograms, but rather as dot plots with selected pairs of markers that illustrate both the shared and distinct populations present for each significant pairwise set of markers.

3. Several FACS stains shown do not appear convincing and the gates used to determine positivity do not always appear appropriate or consistent. For instance, PDGFRb and CD34 staining does not display sufficient signal intensity to allow for meaningful separation versus the FMO control. PDGFRa staining intensity is marginal. If an internal positive control can be demonstrated showing robust positive control staining on an alternative population, this degree of staining would appear to be most consistent with the absence of these markers, not their expression as is claimed in the text. Regarding the gates in the histograms such as Extended Data Fig 2c and 2d, these do not appear to be set in a consistent manner (such as gated such that only a very small and consistent amount of the FMO control falls within each gate), with most gates showing an inappropriately large overlap with the FMO control.

4. In general, the very limited numbers of markers evaluated here and their unclear relevance to functionally defined populations of skeletal progenitors tempers enthusiasm for the flow cytometry data presented. This is emphasized by most of the markers used appearing to not show clear staining. Use of a larger panel, including a greater number of markers used in the Chan et al. Cell 2015 paper, would be helpful for comparison to other literature and thoroughness of characterization, though of secondary importance given the other analysis performed.

5. The term MSC is so poorly defined and a source of such great confusion that it would be better to avoid this terminology altogether, especially in reference to muscle. No assay of stemness of the population studied are provided to justify a stem cell label.

6. Overall, a good amount of detail is provided regarding the methods for the single cell analysis and the methods used appear appropriate. Further clarification on a few points in the methods would be helpful: was there any filtering or analysis of rRNA gene content? Do cell cycle effects appear to drive clustering? If so, was there any regression of cell cycle effects on clustering or other means to account for this? Were the samples all harvested on the same day or on staggered days, and were batch effects excluded as a possible cause of populations differences between timepoints?

7. Very heterogeneous populations of Prx1-lineage positive cells were transplanted as shown by the single cell analysis. Overall, the results of this manuscript would be much stronger if Prx1-lineage cells were fractionated to identify the specific population contributing to fracture callus repair. This issue especially impacts claims such as whether Prx1-lineage progenitors undergo fibrogenesis before

switching to a chondrogenic fate, as this could instead simply reflect the activity of different subpopulations within the Prx1-lineage pool.

8. Impact of the imatinib model is diluted by the lack of supporting studies. It is unclear if the effects of imatinib specifically involve the muscle resident Prx1-lineage cell differentiation pathway described here, especially given that the molecular and cellular targets of imatinib are undefined.

Reviewer #2 (Remarks to the Author):

The authors demonstrate that a subset of muscle cells marked by Prx1-cre expression contribute to fracture healing of the tibia. They also show that these muscle cells change gene expression in response to fracture and that severe muscle damage changes the response to fracture. My concerns primarily involve need for more clarity about their models and their generalizability.

Minor concerns

1. Page 4. What EDL means (extensor digitorum longus) should be spelled out at first use (here). Also spell out what TA means on first use.
2. Extended Data 2: I suspect there's something wrong with the labeling of the clusters for the Chondro-osteogenic markers: 012738656. There are 2 6's and no 4's.

Major concerns

3. In Figure 1, muscle is transplanted away from its natural site, to a site adjacent to a fracture. In that setting, muscle cells marked by Prx1cre invade the fracture callus. A more relevant experiment would have been to transplant the PRx1cre marked muscle into the normal site of the EDL muscle or to replace the muscle near the tibia (replacing or adding to the endogenous EDL muscle of WT mice to see if cells of a muscle in its normal location, might be able to migrate to the site of fracture and participate in callus formation). It looks to me that muscle was put closer to the tibia than normal; normally muscle attaches to bone only through a tendon. Maybe this was done, but it's not obvious from the description.
4. Page 8. The authors show in the graphs in Figure 5e that, after imatinib, fracture sites have less cartilage, bone, and fibrosis. They call this improvement, but do not show evidence (by what criteria?) of improvement in healing/function. I'm not sure what's good about less cartilage and bone. All the authors show is less of everything. Does this lead to a better result in healing and function?
5. P21, description of fracture procedure. The authors describe drilling three holes through the cortex where the fracture is cut. I am not sure the purpose of these three holes. Is that procedure designed to disrupt the periosteum to allow muscle cells to get to the fracture site? Are these three holes crucial for muscle participation in fracture repair? The authors need to clarify this issue.
6. On page 9-10, the authors state, "In bone regeneration, fibrotic progenitors are recruited from adjacent skeletal muscle after polytrauma and cannot be defined as a population of progenitors distinct from skeletal progenitors." I'm not sure that the authors have tried very hard to distinguish these muscle Prx1-marked cells from skeletal Prx1-marked cells. In vitro fates of MSCs is a weak criterion for identity, since lots of cells can be forced into trilineage differentiation. To make the claim here, at a minimum I'd like to see a comparison of the RNAseq clusters marked by Prx1-cre in skeletal tissue and the clusters marked by Prx1-cre in muscle. If those clusters are the same, then the authors' claim would be supported.

Reviewer #3 (Remarks to the Author):

Tissue-resident mesenchymal progenitors (MPs) play fundamental roles in tissue renewal, regeneration and repair. In skeletal muscle, this population is heterogeneous in nature containing at least 3 well-defined populations, fibro-adipogenitors (FAPs), "pericytes" and tenogenic precursors. Following injury-induced activation, these progenitors exit quiescence and produce progeny that contribute to multiple facets of regeneration, with the FAPs exhibiting a more temporary role in muscle regeneration. The manuscript by Anais et al. explores a potential role for muscle-resident MPs in bone regeneration. This concept stems from previous observations related to the impact of severe trauma on the musculoskeletal system and the resulting deficiencies in regeneration. Using a transplant-based model incorporating lineage-tracing capabilities the group shows that muscle-associated MPs participate in bone regeneration, by directly contributing cells to the fracture callus. They suggest that skeletal muscle FAPs are plastic in nature and can produce progeny that contribute to fracture repair. The most exciting component of the paper shows that tissue non-resident (but adjacent) skeletal muscle MPs can potentially participate directly in bone fracture repair. However, to rigorously support this claim a number of controls need to be considered.

Major comments:

- 1) At present well-defined lineage-tracing mouse lines do not exist to differentially label periosteal-MPs versus skeletal muscle MPs, thus, the use of a transplant model is reasonable. However, with the exception of a low-resolution image in Fig. 5a, no images are included prior to fracture. Humphreys and colleagues have shown using a Gli1-based lineage tracing line (Kramann R. et al., 2015, Cell Stem Cell) that MPs will migrate out of bone chips placed in culture. As MPs have a robust response to injury, it is expected following transplantation into a new site that there will be extensive MP mobilization in and around the transplantation site. Characterization of the distribution of transplanted MPs prior to fracture is critical to providing a baseline for understanding their participation in bone repair.
- 2) In Figure 1, lineage-labelled cells contribute to the fracture callus, but at what frequency and is this consistent and robust amongst independent transplant experiments? Following repair, what is the distribution and frequency of labelled osteocytes, strictly periosteal, are labelled-cells observed on the endosteal surface or contribute to endosteal-lining cells?
- 3) On pg. 4, the term osteochondroprogenitors within skeletal muscle is used and these are thought to derive from one or more of the Prx1-expressing MPs in muscle. In models of intramuscular bone formation induced by injection of BMPs or in genetic models of fibrodysplasia ossificans progressiva, muscle-resident MPs directly contribute to bone formation and this has been linked to the Pdgfra+ FAP population and not a resident osteochondroprogenitor population.
- 4) The impact of polytrauma on bone regeneration is challenging to interpret as there is likely a multitude of factors and processes in play, thereby complicating resolution of underlying mediators. Ideally in this scenario it would be useful to manipulate the transplanted MPs post-injury to better understand their participation. This strategy would also aid generally in providing more insights into muscle MP contribution to bone healing. For these purposes, it would be useful to incorporate a selective ablation model whereby iDTR is expressed in the transplanted MPs and DTA fragment administration can be used to selectively ablate the transplanted MPs. This would answer a critical question with respect to whether muscle-resident MPs are necessary in bone repair and if ablation of the transplanted MPs leads to reduced fracture repair in both models. There may be other ways to address this issue.
- 5) The concept that the D5-polytrauma samples exhibit a delay in their activation post-injury is not well developed. If this is the case, then it would be expected that these cells would exhibit different kinetics with respect to proliferation. This could be substantiated by directly comparing EdU

incorporation or similar in the transplanted MPs and resident MPs (along with appropriate surface markers) between the two models. Based on data presented in extended data Fig. 5 there doesn't appear to be an appreciable change in cell proliferation in the different models.

6) In Fig. 2g, it is somewhat surprising that new (potentially temporary) clusters do not emerge as injury-induced activation is typically associated with large changes in the transcriptome resulting in the appearance of new clusters. In this regard, only clusters 0 and 5 seem to change markedly over time. A comparison to untransplanted "normal" MPs would be helpful.

7) In the polytrauma model is there an increase in cell death associated with this injury that could contribute to failed/reduced repair.

8) Imatinib treatment will impact both transplanted and non-transplanted MPs, and thus it is difficult to ascertain if these treatments act through sole modulation of the transplanted muscle MPs. Furthermore, benefit in fracture repair would need to be assessed using quantitative measures of fracture healing such as bone histomorphometric analysis. Furthermore, if this is the case, then Imatinib treatment should improve bone healing following polytrauma in non-manipulated animals.

9) Ideally for these types of experiments transplants would have been placed into mice that would also allow for lineage-tracing of host MPs such that direct comparisons could be made with regards to host vs. transplant MP contribution to the fracture callus and bone repair.

Minor comments:

1) the markers used in Table 1 should be adjusted as they appear in other populations of interest. For instance, in addition to FAPs, Tagln is also expressed in VSMCs and pericytes, Pth1r is expressed in MPs as is Vegfa and Wnt5a. For the chondrogenic compartment, Sox9 (not perfect, also in MPs, but enriched in chondrocytes), Sox6, Col2a1 and Acan as suggested are reasonable markers.

2) To consider, following TAM injections, a longer washout period should be used of 2 weeks, but ideally a month.

We thank the reviewers for their time and insightful comments. We provide below a point-by-point response to the comments and have incorporated significant changes in the manuscript along with additional data in response to these comments.

Response to reviewers:

Reviewer #1 (Remarks to the Author):

In this manuscript, Anais et al. evaluate the contribution of muscle-resident Prx1-lineage cells to fracture repair. Overall the topic of which cells mediate fracture repair is of interest, especially given our increasingly granular understanding of skeletal populations. This manuscript has several methodological and conceptual strengths that are worth noting. First, consideration of combined bone/muscle injury and the mechanisms underlying why outcomes after this may diverge from bone injury only is a novel concept within the mechanistic bone biology literature. It is likely that this concept and the accompanying model developed here will be of broader relevance and interest for the field. There are also some methodologic strengths to note. First, there is an attempt to comprehensively characterize the muscle resident PRX1-lineage cells, though there are a number of specific concerns with elements of this approach as described below. Second, the use of multiple experimental timepoints for scRNA-seq studies strengthens the pseudotime analysis performed. Lastly, a transplantation system is used that provides strong evidence that the PRX1-lineage cells tracked are truly an extraskeletal resident population. Additionally, the identification of a possible source for fibrotic tissue present in pseudoarthroses/non-unions is of interest and has potential clinical relevance.

Despite these strengths, a number of important weaknesses are noted that will be critical to address. Chief among these is 1) that the evidence for the contribution of the muscle resident Prx1-lineage cells to fracture healing is not currently convincing. Second, the outcome measures from fracture models utilized are mostly histology based, whereas uCT or other methods provide more robust and convincing data. All of these issues are described in detail below.

Major points:

1. The most important overall limitation in the manuscript is that the evidence that Prx1-lineage muscle resident cells have an important contribution to fracture healing is currently not convincing. The key experiment to address this central point of the manuscript is in Fig 1c (and extended fig 1c). This shows that the GFP+ Prx1-lineage cells appear to be largely restricted to peripheral portions of the callus region at days 7 and 14. From the images provided, it is unclear if the Fig 1c GFP+ cells are truly in the fracture callus or largely restricted to peripheral connective tissue. Additionally, none of the GFP+ cells in the day 14 samples (Fig 1c) show convincing chondrocyte morphology, and no co-labeling with cartilage markers is provided. No quantitative measure of the relative amounts of each of these cellular compartments that are reconstituted with Prx1-lineage cells is provided. The pictures presented give the impression that much of this engraftment is transient given that what appears to be a single cell is all that is shown in the day 28 image. This single cell at day 28 is not convincing for contribution to osteoblast lineage cells. Overall, the use of a great number of high-power inserts throughout Fig 1 without reference to a lower power image (with the exception of Fig 1b) lessens the impact of this data. More direct clarification of whether a contribution to osteogenic lineages is claimed would be helpful, as well as a comparative discussion of the results of Fig 1a vs Fig 1b vs Fig 1c. From the model presented, one would expect fig 1c to give the same results as Figs 1a and b, but the data shown gives a different impression, especially as regards the contribution to chondrogenesis.

Importantly, there is no direct functional evidence for the contribution of Prx1-positive cells to fracture healing and claims about the role of muscle resident Prx1-positive cells are much weaker without supporting functional data.

Response: We have re-organized figure 1 and added data in Supplementary Figures 1-3 to provide strong evidence that Prx1-derived muscle cells directly contribute to fracture healing. Figure 1 now allows a better visualization of Tomato-labeled or GFP-labeled muscle-derived cells migrating in the center of the fracture callus, demonstrating that the contribution is not restricted to peripheral portions of the callus. We show low magnification images with clear delimitation of the callus, as well as high magnification images relative to the low magnification images. High magnification of cartilage and bone better illustrate cellular morphology and double immunostaining with chondrocyte and osteoblast markers in Figure 1b. Immunostaining provides more convincing evidence of muscle contribution to cartilage (Colx+ cells) and bone (Osx+ cells). Further, we illustrate the timing of Prx1+ cell activation and migration within the callus in Supplementary Figure 3a. We also performed quantification of GFP+ muscle-derived cells within fibrous tissue, cartilage, endochondral bone and intramembranous bone showing that muscle-derived cells mostly contribute to endochondral ossification (supplementary figure 1b).

Fig. 1 shows that GFP+ cells colocalized with Osx+ cells, demonstrating that Prx1-derived skeletal muscle cells are able to contribute to the osteoblastic lineage. However, since muscle-derived cells contribute mainly to the endochondral ossification process, this contribution is not persistent due to the dynamic process of cartilage replacement by bone, and bone remodeling in the center of the callus. This endochondral ossification process, although transient, is required for fracture consolidation as shown in our recent publication (Julien et al., Stem Cell Reports, 2020 PMID32916123), which does not undermine the role of muscle-derived cells for bone healing.

As shown by our single cell transcriptomic and RTqPCR results, most markers previously identified in periosteum or bone marrow cells are also expressed in skeletal muscle progenitors. In the absence of specific marker to target this muscle-derived population, functional studies based on genetic ablation or gene targeting cannot be employed. For this reason, we developed transplantation approaches, and assessed the functional consequences of polytrauma on skeletal muscle mesenchymal progenitors and on bone repair. Local induction protocols to target muscle specifically can also be envisioned but do not currently provide sufficient recombination efficiency and will require considerable optimization in the future.

2. A major limitation of the fracture studies is that primarily histologic endpoints are used. While histology is essential, especially for evaluating fibrous and cartilaginous callus tissue, uCT is still needed, as this is the most technically robust method to assess for overall delays in bone formation and for lack of bone bridging/non-union phenotypes. This issue is especially important as the non-stabilized fracture model utilized here has a very high degree of variability. Thus, robust quantitative measures (such as absolute measures of callus mineralization or uCT determined bridging/non-union rates) needed to ensure that variability in the model is accounted for. Histomorphometry does not provide as convincing data due to the sampling issues inherent in histologic approaches. uCT does not need to be universally applied, but key results should be confirmed by targeted uCT analysis, such as they key groups used to argue for altered overall skeletal healing in the polytrauma and imatinib models.

Response: We now provide microCT analyses as a more robust assessment of delayed healing/non-union in the semi-stabilized and non-stabilized fracture models following musculoskeletal trauma (Figure 4 and supplementary figure 8). We also assessed healing using microCT analyses in the semi-stabilized model to show more convincingly the positive impact of Imatinib treatment on bone repair (Figure 7).

3. A control, or at least reference to literature examining this point, showing that Prx1 cre is not deleting in myogenic lineages in the initial donors is needed.

Response: The myogenic lineage in the limb is strictly derived from Pax3/Pax7 lineage (Relaix et al., Nature, 2005). Supplementary Figure 2 shows that Pax7-labeled myogenic lineage does not contribute to callus formation. To confirm that myogenic and Prx1 mesenchymal lineages define distinct cell populations within skeletal muscle, we performed additional scRNAseq analyses and

show absence of myogenic marker expression in any of the clusters identified in Prx1-sorted cells (Supplementary Figure 4a). Further, we compared our scRNAseq data with published scRNAseq data of skeletal muscle, and found no overlap between Prx1-derived skeletal muscle cells and myogenic clusters (Supplementary Figure 5).

4. The choice of the unstabilized fracture model is unusual, as stabilized models are usually selected due to both the lesser degree of variation present and, arguably, due to stabilized fractures being a more overall clinically relevant model. Commentary on the rationale for selecting an unstabilized model would be helpful. Given the use of an unstabilized model, there is a concern that the results of the polytrauma model could be entirely due to impairing the ability of surrounding muscle to stabilize the fracture, and not due to any specific impact of the muscle trauma on muscle resident mesenchymal progenitors. Changes in the degree of fracture site stabilization can have very large impacts on outcomes in fracture models, as often impacts from changes in the degree of stabilization that are bigger than the impact of biologic factors under study.

Response: We show that skeletal muscle contribution to bone repair is mainly through the endochondral ossification process. This process is amplified in a non-stabilized fracture model. Therefore this model is preferred as an experimental model to study the role of muscle in bone repair. However, we now report additional data to show that the role of skeletal muscle in bone repair is confirmed when using a semi-stabilized fracture model. In the semi-stabilized model, we also observed cellular contribution of skeletal muscle (Supplementary Figure 8), delayed healing following polytrauma (Supplementary Figure 8), and impact of Imatinib treatment in correcting the polytrauma phenotype (Figure 7). We have added a comment in the results section and in the methods to justify these 2 models.

Minor points

1. The term EDL (extensor digitorum longus?) should be defined upon first use.

Response: We defined EDL (*Extensor Digitus Lengus*) upon first use.

2. FACS plots should not be displayed as histograms, but rather as dot plots with selected pairs of markers that illustrate both the shared and distinct populations present for each significant pairwise set of markers.

Response: We changed the representation of FACS plots and show dot plots in Supplementary Figure 4b-d.

3. Several FACS stains shown do not appear convincing and the gates used to determine positivity do not always appear appropriate or consistent. For instance, PDGFRb and CD34 staining does not display sufficient signal intensity to allow for meaningful separation versus the FMO control. PDGFRa staining intensity is marginal. If an internal positive control can be demonstrated showing robust positive control staining on an alternative population, this degree of staining would appear to be most consistent with the absence of these markers, not their expression as is claimed in the text. Regarding the gates in the histograms such as Extended Data Fig 2c and 2d, these do not appear to be set in a consistent manner (such as gated such that only a very small and consistent amount of the FMO control falls within each gate), with most gates showing an inappropriately large overlap with the FMO control.

Response: FACS analyses have been re-evaluated and new dot plot representation better illustrate positive and negative populations in Supplementary Figure 4b-d.

4. In general, the very limited numbers of markers evaluated here and their unclear relevance to functionally defined populations of skeletal progenitors' tempers enthusiasm for the flow cytometry data presented. This is emphasized by most of the markers used appearing to not show clear staining. Use of a larger panel, including a greater number of markers used in the Chan et al. Cell 2015 paper, would be helpful for comparison to other literature and thoroughness of

characterization, though of secondary importance given the other analysis performed.

Response: FACS analyses aimed to define the Prx1-derived skeletal muscle population relative to cell populations marked by Pdgfra, Sca1 and Cd34 previously described in skeletal muscle (Uezumi A, Nat. Cell. Biol., 2010 & Joe AWB, Nat. Cell. Biol., 2009). We now show additional scRNAseq analyses to compare the identity of Prx1-derived skeletal muscle mesenchymal progenitors with skeletal/stem progenitor cells previously characterized in bone by (Chan et al., Cell, 2015; Debnath, 2018; Zhou, 2014; Worthley, 2015; Mendez-Ferrer, 2010; Park, 2012) in Supplementary Figure 4e-f. These additional analyses now provide a more thorough evaluation of Prx1-derived skeletal muscle mesenchymal progenitors and show that this population from skeletal muscle expresses common markers with skeletal stem/progenitor cells from bone.

5. The term MSC is so poorly defined and a source of such great confusion that it would be better to avoid this terminology altogether, especially in reference to muscle. No assay of stemness of the population studied are provided to justify a stem cell label.

Response: We have removed the term MSC and now refer to Mesenchymal Progenitor (MP).

6. Overall, a good amount of detail is provided regarding the methods for the single cell analysis and the methods used appear appropriate. Further clarification on a few points in the methods would be helpful: was there any filtering or analysis of rRNA gene content? Do cell cycle effects appear to drive clustering? If so, was there any regression of cell cycle effects on clustering or other means to account for this? Were the samples all harvested on the same day or on staggered days, and were batch effects excluded as a possible cause of population differences between timepoints?

Response: We did not use filtering based on rRNA content, as our dataset did not present ribosomal genes as highly variable features. We performed clustering taking in account cell cycle state and we observed similar results. Therefore cell cycle effects did not drive clustering (see Supplementary Figure 9c, d). Samples were harvested on two different days. According to news technics of data integration developed by Satija Lab and described in (Stuart T., Cell, 2019), we were able to integrate different scRNAseq datasets to perform subsequent analyses with limited batch effect causing population differences.

7. Very heterogeneous populations of Prx1-lineage positive cells were transplanted as shown by the single cell analysis. Overall, the results of this manuscript would be much stronger if Prx1-lineage cells were fractionated to identify the specific population contributing to fracture callus repair. This issue especially impacts claims such as whether Prx1-lineage progenitors undergo fibrogenesis before switching to a chondrogenic fate, as this could instead simply reflect the activity of different subpopulations within the Prx1-lineage pool.

Response: Given the in vivo potential of the Prx1-population to contribute to fibrogenesis and chondrogenesis, we performed scRNAseq at different time points to specifically determine whether a sub-population undergoes fibrogenesis and another one chondrogenesis. Results in Fig. 3 and Supplementary Figure 6 show that although the Prx1-derived population is heterogeneous, we could not identify a fibrogenic cluster and a chondrogenic cluster within the Prx1-derived muscle population. Although this population is heterogeneous, the scRNAseq data show that the entire population responds to bone fracture by undergoing fibrogenesis before chondrogenesis.

8. Impact of the imatinib model is diluted by the lack of supporting studies. It is unclear if the effects of imatinib specifically involve the muscle resident Prx1-lineage cell differentiation pathway described here, especially given that the molecular and cellular targets of imatinib are undefined.

Response: Imatinib targets Pdgfra, Bcr-Abl, c-Kit, and is used for cancer treatment, specifically in chronic myelogenous leukaemia (Capdeville R, Nature Reviews Drug Discovery, 2002, PMID: 12120256). Pdgfra signaling is also known to play a central role in fibrotic processes. Indeed, Imatinib have been used as anti-fibrotic treatment in experimental models (Daniels CE, JCI, 2004,

PMID: 15520863; Huang P, FASEB Journal, 2016, PMID: 26683699). We show that Prx1-derived cells in skeletal muscle are the main contributor to callus fibrosis and express *Pdgfra* (see Figure 2d and Supplementary Figure 4c). Thus, Imatinib appeared as a good candidate to reduce callus fibrosis and improve bone healing after polytrauma.

Reviewer #2 (Remarks to the Author):

The authors demonstrate that a subset of muscle cells marked by Prx1-cre expression contribute to fracture healing of the tibia. They also show that these muscle cells change gene expression in response to fracture and that severe muscle damage changes the response to fracture. My concerns primarily involve need for more clarity about their models and their generalizability.

Minor concerns

1. Page 4. What EDL means (extensor digitorum longus) should be spelled out at first use (here). Also spell out what TA means on first use.

Response: We now define EDL (*Extensor Digitus Lengus*) and TA (*tibialis anterior*) upon first use.

2. Extended Data 2: I suspect there's something wrong with the labeling of the clusters for the Chondro-osteogenic markers: 012738656. There are 2 6's and no 4's.

Response: We have corrected this mistake.

Major concerns

3. In Figure 1, muscle is transplanted away from its natural site, to a site adjacent to a fracture. In that setting, muscle cells marked by Prx1cre invade the fracture callus. A more relevant experiment would have been to transplant the PRx1cre marked muscle into the normal site of the EDL muscle or to replace the muscle near the tibia (replacing or adding to the endogenous EDL muscle of WT mice to see if cells of a muscle in its normal location, might be able to migrate to the site of fracture and participate in callus formation). It looks to me that muscle was put closer to the tibia than normal; normally muscle attaches to bone only through a tendon. Maybe this was done, but it's not obvious from the description.

Response: We transplanted EDL muscle by suturing tendon-to-tendon. We have clarified this in the methods. The removal of endogenous skeletal muscle to replace with the graft would lead to massive injury and artificial perturbations of the fracture environment, which would have an impact of repair. Therefore, we transplanted EDL muscle without disrupting the endogenous surrounding muscle in order to better evaluate the endogenous contribution.

We chose EDL muscle because it is easier to perform tendon-to-tendon grafting. The EDL is transplanted closer to the bone than the endogenous EDL. However, when we transplanted TA, which is in closer contact with tibia, we also detected contribution. Therefore, the contribution was not due to an artificial response because of closer contact between grafted muscle and bone.

We also observed that when EDL muscle graft was not sutured correctly, the contribution was minimal, certainly due to the distance between the grafted muscle and the bone, suggesting that muscles in close proximity to the bone are more likely to contribute to repair than muscles away from the bone.

4. Page 8. The authors show in the graphs in Figure 5e that, after imatinib, fracture sites have less cartilage, bone, and fibrosis. They call this improvement, but do not show evidence (by what criteria?) of improvement in healing/function. I'm not sure what's good about less cartilage and bone. All the authors show is less of everything. Does this lead to a better result in healing and function?

Response: Polytrauma causes delayed callus, cartilage and bone resorption, as well as delayed

fibrotic remodeling. This delay results in increased callus, cartilage and bone volumes, as well as increased fibrosis at late stages of repair (see Fig. 4). In Imatinib treated mice, we observed a decrease in callus, cartilage, bone and fibrosis at d21 compared to PBS, suggesting an improved resorption. This effect was indicative of improved healing. We now we used a semi-stabilized fracture model (Fig. 7 and Supp. Fig. 8) and microCT analyses to better assess the effect of polytrauma and Imatinib on bone repair. We used a scoring system to evaluate fracture consolidation and show that Imatinib treatment can rescue the phenotype in 50% of cases. This is correlated with decrease fibrosis at d21 (see Fig. 7).

5. P21, description of fracture procedure. The authors describe drilling three holes through the cortex where the fracture is cut. I am not sure the purpose of these three holes. Is that procedure designed to disrupt the periosteum to allow muscle cells to get to the fracture site? Are these three holes crucial for muscle participation in fracture repair? The authors need to clarify this issue.

Response: To clarify the methods, we indicate that we performed osteotomy by cutting the bone. Drilling holes allows to perform a clean cut without causing damage to the bone cortex. This procedure does not disrupt the periosteum or does not allow muscle cells to get into the fracture site. We removed the mention “drilling three holes” from the method to avoid confusion.

6. On page 9-10, the authors state, “In bone regeneration, fibrotic progenitors are recruited from adjacent skeletal muscle after polytrauma and cannot be defined as a population of progenitors distinct from skeletal progenitors.” I’m not sure that the authors have tried very hard to distinguish these muscle Prx1-marked cells from skeletal Prx—marked cells. In vitro fates of MSCs is a weak criterion for identity, since lots of cells can be forced into trilineage differentiation. To make the claim here, at a minimum I’d like to see a comparison of the RNAseq clusters marked by Prx1-cre in skeletal tissue and the clusters marked by Prx1-cre in muscle. If those clusters are the same, then the authors’ claim would be supported.

Response: We have modified this statement that was misleading:

“In bone regeneration, fibrotic progenitors are recruited from adjacent skeletal muscle after polytrauma and ~~cannot be defined as a population of progenitors distinct from skeletal progenitors.~~” was changed to “In bone regeneration, fibrotic progenitors are recruited from adjacent skeletal muscle after polytrauma and are derived from the same pool of progenitors that will undergo chondrogenesis”

In this part of the discussion, we did not mean to refer to the skeletal progenitors from bone. We are referring to the heterogeneity of the skeletal muscle cell population. ScRNAseq analyses revealed cellular heterogeneity but did not demonstrate the presence of separate pools of progenitors that give rise either to fibrosis or to chondrocytes. Instead, the entire FAP/MP population can undergo fibrosis prior to chondrogenesis (see also response to reviewer1 #7). Deregulation of this initial response leads to abnormal chondrogenesis and fibrotic accumulation. However, in response to the reviewer’s comment regarding the comparison of Prx1-derived cells in muscle with cells from skeletal tissues, we have performed analyses of skeletal/stem progenitor cell markers in Supplementary Figure 4e-f (see also response to Reviewer 1, minor points comment #4).

Reviewer #3 (Remarks to the Author):

Tissue-resident mesenchymal progenitors (MPs) play fundamental roles in tissue renewal, regeneration and repair. In skeletal muscle, this population is heterogeneous in nature containing at least 3 well-defined populations, fibro-adipoprogenitors (FAPs), “pericytes” and tenogenic precursors. Following injury-induced activation, these progenitors exit quiescence and produce progeny that contribute to multiple facets of regeneration, with the FAPs exhibiting a more temporary role in muscle regeneration. The manuscript by Anais et al. explores a potential role for

muscle-resident MPs in bone regeneration. This concept stems from previous observations related to the impact of severe trauma on the musculoskeletal system and the resulting deficiencies in regeneration. Using a transplant-based model incorporating lineage-tracing capabilities the group shows that muscle-associated MPs participate in bone regeneration, by directly contributing cells to the fracture callus. They suggest that skeletal muscle FAPs are plastic in nature and can produce progeny that contribute to fracture repair. The most exciting component of the paper shows that tissue non-resident (but adjacent) skeletal muscle MPs can potentially participate directly in bone fracture repair. However, to rigorously support this claim a number of controls need to be considered.

Major comments:

1) At present well-defined lineage-tracing mouse lines do not exist to differentially label periosteal-MPs versus skeletal muscle MPs, thus, the use of a transplant model is reasonable. However, with the exception of a low-resolution image in Fig. 5a, no images are included prior to fracture. Humphreys and colleagues have shown using a Gli1-based lineage tracing line (Kramann R. et al., 2015, Cell Stem Cell) that MPs will migrate out of bone chips placed in culture. As MPs have a robust response to injury, it is expected following transplantation into a new site that there will be extensive MP mobilization in and around the transplantation site. Characterization of the distribution of transplanted MPs prior to fracture is critical to providing a baseline for understanding their participation in bone repair.

Response: In Fig. 2a, we added a lower magnification of Prx1Cre;mTmG TA muscle to allow a better visualization of Prx1-derived cells within intact skeletal muscle.

In order to demonstrate that the migration of Prx1-derived muscle cells occurs specifically in response to fracture, we compared the distribution of Prx1-derived muscle cells after EDL transplantation adjacent to an intact tibia and a fractured tibia (see Supplementary Fig. 3a, b). At day 5 post-transplantation, Prx1-derived cells are restricted to the grafted muscle in the absence of bone fracture. No GFP+ derived cells are observed within adjacent endogenous muscle, periosteum or bone marrow. However, when EDL muscle was transplanted adjacent to a fracture, GFP+ cells are found outside the transplanted EDL at the fracture site. By d21 post-transplantation without fracture, GFP+ cells are rare in the grafted muscle and undetectable in surrounding tissues. After fracture, a strong GFP+ signal is detected by d7 and 14 in the fracture callus, and is still detected by d21. In the absence of bone fracture, EDL muscle transplantation leads to activation of Prx1-derived cells within the grafted muscle, without migration in surrounding tissues. Therefore, migration of Prx1-derived cells within the fracture callus occurs specifically in response to fracture.

2) In Figure 1, lineage-labelled cells contribute to the fracture callus, but at what frequency and is this consistent and robust amongst independent transplant experiments? Following repair, what is the distribution and frequency of labelled osteocytes, strictly periosteal, are labelled-cells observed on the endosteal surface or contribute to endosteal-lining cells?

Response: We have quantified the EDL muscle derived GFP+ cells in fibrous tissue, cartilage, endochondral bone and intramembranous bone. We show that quantification is consistent across samples (see Supplementary Figure 1b). We did not detect GFP+ cells within periosteum, bone marrow compartment or at the endosteal surface. The GFP+ osteocytes are localized in the center of the callus where endochondral ossification occurs. In this region bone becomes largely remodeled during the course of repair and muscle-derived osteoblasts are rare by day 28. These quantitative analyses further reinforce that skeletal muscle mainly contributes to the endochondral ossification process during bone repair.

3) On pg. 4, the term osteochondroprogenitors within skeletal muscle is used and these are thought to derive from one or more of the Prx1-expressing MPs in muscle. In models of intramuscular bone formation induced by injection of BMPs or in genetic models of fibrodysplasia ossificans progressiva, muscle-resident MPs directly contribute to bone formation and this has

been linked to the Pdgfra+ FAP population and not a resident osteochondroprogenitor population.

Response: To avoid confusion, we now refer to the MPs from skeletal muscle that have the potential to give rise to chondrocytes and osteoblasts during bone repair. We refer to “osteochondroprogenitor” only when assessing the population defined by Chan et al. (see Supplementary Figure 4). ScRNAseq analyses show that Prx1-derived MPs give rise to fibrogenesis and chondrogenesis. In Supplementary Fig. 5, we showed that this Prx1-marked population overlaps with the FAP population defined by the markers Sca1, Cd34 and Pdgfra (Uezumi A, 2010; Joe, 2009) and other MP clusters defined by (Scott et al., 2019; Tabula Muris, 2018; Giordanni et al., 2019). Our result thus correlate with previously published work showing that muscle-resident MPs can contribute the bone formation in FOP and are linked to Pdgfra+ FAP population.

4) The impact of polytrauma on bone regeneration is challenging to interpret as there is likely a multitude of factors and processes in play, thereby complicating resolution of underlying mediators. Ideally in this scenario it would be useful to manipulate the transplanted MPs post-injury to better understand their participation. This strategy would also aid generally in providing more insights into muscle MP contribution to bone healing. For these purposes, it would be useful to incorporate a selective ablation model whereby iDTR is expressed in the transplanted MPs and DTA fragment administration can be used to selectively ablate the transplanted MPs. This would answer a critical question with respect to whether muscle-resident MPs are necessary in bone repair and if ablation of the transplanted MPs leads to reduced fracture repair in both models. There may be other ways to address this issue.

Response: In order to determine the role of skeletal muscle derived cells during bone repair, we performed EDL muscle transplantation in fracture and polytrauma and showed that the contribution of EDL-derived cells to the cartilage is decrease in the context of polytrauma (see Fig. 3). To better understand how polytrauma environment affect MP from skeletal muscle, we isolated MP from fracture and polytrauma to compare their molecular response using scRNAseq technology. We showed that skeletal muscle injury impact cell metabolism and ECM secretion at d3 and d5 (see Fig. 5).

A depletion approach in the EDL graft would likely not provide a phenotype, as the endogenous muscle cells would not be targeted. As indicated in response to Reviewer 1, a genetic approach would require identification of a specific marker for the skeletal muscle MP population distinct from bone. Our results show that skeletal muscle MP and skeletal stem/progenitors from bone express common markers. Local induction protocols to target muscle specifically can also be envisioned but do not currently provide sufficient recombination efficiency and will require considerable optimization in the future. In the context of this study, the transplantation approaches were the most informative to understand the impact of polytrauma on the cellular contribution of skeletal muscle. We do not exclude other factors that may have an impact on the polytrauma phenotype. However, our results clearly highlight the role of skeletal muscle in producing persistent fibrosis in this context.

5) The concept that the D5-polytrauma samples exhibit a delay in their activation post-injury is not well developed. If this is the case, then it would be expected that these cells would exhibit different kinetics with respect to proliferation. This could be substantiated by directly comparing EdU incorporation or similar in the transplanted MPs and resident MPs (along with appropriate surface markers) between the two models. Based on data presented in extended data Fig. 5 there doesn't appear to an appreciable change in cell proliferation in the different models.

Response: The scRNAseq analyses show indeed that the defective activation of muscle MPs in polytrauma was not correlated with changes in proliferation, rather abnormal fibrotic response. In addition, when we quantified the GFP+ signal, we did not detect changes in the expansion of the MP in skeletal muscle surrounding the fracture after polytrauma compared to fracture alone (Figure 6). Therefore, we did not perform further analyses of proliferation (nor cell death as indicated below

in comment #7). Instead, we concentrated on the changes in the fibrogenic response that are more reflective of the defective activation of MP after polytrauma and precede the deficient chondrogenic response. These scRNAseq analyses correlate with the phenotypic analyses.

6) In Fig. 2g, it is somewhat surprising that new (potentially temporary) clusters do not emerge as injury-induced activation is typically associated with large changes in the transcriptome resulting in the appearance of new clusters. In this regard, only clusters 0 and 5 seem to change markedly over time. A comparison to untransplanted “normal” MPs would be helpful.

Response: In these experiments, we did not analyze transplanted MPs. We directly isolated MP (Prx1+ derived cells based on GFP expression) from freshly dissected muscle next to the fractured tibia to analyze their response to fracture or polytrauma. We included a day 0-uninjured control group in order to analyze the response to fracture or polytrauma (Figures 3 and 5).

7) In the polytrauma model is there an increase in cell death associated with this injury that could contribute to failed/reduced repair.

Response: As observed for proliferation, we did not detect changes in cell death as shown in (Figure 5d).

8) Imatinib treatment will impact both transplanted and non-transplanted MPs, and thus it is difficult to ascertain if these treatments act through sole modulation of the transplanted muscle MPs. Furthermore, benefit in fracture repair would need to be assessed using quantitative measures of fracture healing such as bone histomorphometric analysis. Furthermore, if this is the case, then Imatinib treatment should improve bone healing following polytrauma in non-manipulated animals.

Response: We did not perform transplantations in this experiment. We assessed the overall impact of Imatinib treatment on bone repair. We assessed via histomorphometric analyses in non-stabilized and semi-stabilized fracture models (Figure 7 and Supplementary Fig. 10). We also performed quantitative measurements using microCT analyses in the semi-stabilized fracture model to show improved healing in the treated group (Figure 7)(see also response reviewer1 comments#2&4 and reviewer2 comment#4).

9) Ideally for these types of experiments transplants would have been placed into mice that would also allow for lineage-tracing of host MPs such that direct comparisons could be made with regards to host vs. transplant MP contribution to the fracture callus and bone repair.

Response: Currently, there are no marker to distinguish Prx1-populations from periosteum, bone marrow and muscle. If we label MP in the host, we would also label periosteum and bone marrow-derived cells. Therefore, transplantation approaches are necessary to trace donor-derived muscle cells specifically.

Minor comments:

1) the markers used in Table 1 should be adjusted as they appear in other populations of interest. For instance, in addition to FAPs, Tagln is also expressed in VSMCs and pericytes, Pth1r is expressed in MPs as is Vegfa and Wnt5a. For the chondrogenic compartment, Sox9 (not perfect, also in MPs, but enriched in chondrocytes), Sox6, Col2a1 and Acan as suggested are reasonable markers.

Response: As there are no strict specific markers for each cell type or state of differentiation, we used a combination of markers according to the literature. To avoid bias in our analysis, we changed the list of genes for the chondrogenic and fibrogenic state. We used the GO list number 0043062 for the fibrogenesis and the GO list number 0051216 for the chondrogenesis.

2) To consider, following TAM injections, a longer washout period should be used of 2 weeks, but ideally a month.

Response: For our purpose, TAM injections are performed 1-week prior analyses (Abou-Khalil R, Stem Cells, 2015). Since we did not see contribution, a longer washout period is not likely to show different results.

REVIEWERS' COMMENTS

Reviewer #1 (Remarks to the Author):

Overall, the revised manuscript adequately addresses concerns raised in response to the initial submission. The figures added to the revised figure 1 are greatly improved and now provide a much clearer picture of the integration of the graft cells to the fracture callus. The use of the semi-stabilized fracture model adequately addresses concerns about secondary effects of polytrauma on fracture stabilization. While questions remain about secondary effects of muscle trauma, this is a reasonable approach given experimental limitations. FACS plots in Fig 4 also appear to be improved in quality. There are a few minor remaining issues noted below, but it is not anticipated that these would require additional experiments. While important questions remain about the data and the concepts introduced here, this manuscript makes a convincing case that muscle resident PRX1-lineage cells can integrate into the fracture callus, setting the stage for future functional characterization of the contribution of these cells and their fractionation into discrete populations. Additionally, the formal consideration of polytrauma performed here also introduces an important concept that has largely not been considered in the basic science skeletal healing literature. The method of muscle transplantation employed here is also novel in this context. Thus, this manuscript makes several clear conceptual contributions to the bone healing literature and will be of great interest to the readership of Nature Communications.

Minor points

1. There are concerns with the analysis added in Sup Fig 4e and portions of the analysis in Sup Fig 4f. While it is appreciated that this was added in response to reviewer comments, the idea that flow cytometry based cellular definitions can be transferred directly into single cell sequencing to identify corresponding cellular populations is not sound. First, scRNA seq only provides a sparse sampling of the transcriptome of each cell. Thus, while the absence of marker expression is fairly definitive by flow cytometry, it is not definitive by single cell RNA sequencing and may simply reflect the low sensitivity of this technique. Second, it is well known that many transcripts display extremely poor correlation to the amount of the corresponding protein when assayed at the single cell level. CTSK expression in Fig 4f may reflect the known expression of CTSK in tendon cells and not necessarily indicate the presence of CTSK-lineage periosteal cells in the pool of cells analyzed. The discordance between LEPR and CXCL12 in the Fig 4f analysis is puzzling given that these markers (LEPR+ vs CXCL12+ "CAR" cells) have been largely shown to refer to the same population (e.g., Matsushita et al. Nat Commun 2020). Perhaps the most likely explanation is that the CXCL12 cells analyzed here do not represent the well-studied CXCL12 abundant reticular cells present in marrow stroma that co-express LEPR, but that these cells are other CXCL12+ populations. Thus, without further qualification or supporting studies, the unstated implication here that CXCL12 positivity relates to well studied CXCL12+ mesenchymal populations in bone may be misleading. Overall it is recommended that this analysis be carefully reconsidered and that the less reliable portions of this analysis be either clearly qualified or omitted.
2. It is recommended that the bone volume in revised Fig 4c be displayed in units corresponding to actual voxel size (e.g., mm³).
3. As the authors note, imatinib has many targets as a fairly nonspecific kinase inhibitor with well known pleiotropic effects. Some acknowledgement of this limitation in the text is needed to balance the statement that it inhibits PDGFR α . Additionally, qualification is needed to acknowledge that imatinib may impact many cell types present in addition to PRX1-lineage muscle resident cells.
4. The bar graph provided in Fig 4c indicating that all of the fractures in both polytrauma and control groups "healed" appears to be at odds with the statement in the text and the left panels of Fig 4c noting an absence of bone bridging in 100% of polytrauma mice. It should be clarified what "healed" means here and how this fits with the non-union/non-bridging seen in polytrauma.

Reviewer #2 (Remarks to the Author):

The authors have improved the manuscript substantially and have addressed all my concerns satisfactorily.

Reviewer #3 (Remarks to the Author):

The authors have adequately addressed my concerns.

Just one point of clarification regarding the ablation experiment. The goal here would be to graft the EDL from a mouse in which resident MPs express iDTR (or a TAM-inducible DTA) into a non-transgenic mouse. Once grafted into a recipient mouse, presumably the transplanted MPs within the EDL could be selectively ablated using TAM or the DTA fragment. A *Pdgfra-CreERT2* mouse would work well for this strategy and has been successfully used by others to ablate muscle-associated MPs (Wosczyzna et al., 2019, Cell Reports).

Reviewer #1 (Remarks to the Author):

Overall, the revised manuscript adequately addresses concerns raised in response to the initial submission. The figures added to the revised figure 1 are greatly improved and now provide a much clearer picture of the integration of the graft cells to the fracture callus. The use of the semi-stabilized fracture model adequately addresses concerns about secondary effects of polytrauma on fracture stabilization. While questions remain about secondary effects of muscle trauma, this is a reasonable approach given experimental limitations. FACS plots in Fig 4 also appear to be improved in quality. There are a few minor remaining issues noted below, but it is not anticipated that these would require additional experiments. While important questions remain about the data and the concepts introduced here, this manuscript makes a convincing case that muscle resident PRX1-lineage cells can integrate into the fracture callus, setting the stage for future functional characterization of the contribution of these cells and their fractionation into discrete populations. Additionally, the formal consideration of polytrauma performed here also introduces an important concept that has largely not been considered in the basic science skeletal healing literature. The method of muscle transplantation employed here is also novel in this context. Thus, this manuscript makes several clear conceptual contributions to the bone healing literature and will be of great interest to the readership of Nature Communications.

Minor points

1. There are concerns with the analysis added in Sup Fig 4e and portions of the analysis in Sup Fig 4f. While it is appreciated that this was added in response to reviewer comments, the idea that flow cytometry based cellular definitions can be transferred directly into single cell sequencing to identify corresponding cellular populations is not sound. First, scRNA seq only provides a sparse sampling of the transcriptome of each cell. Thus, while the absence of marker expression is fairly definitive by flow cytometry, it is not definitive by single cell RNA sequencing and may simply reflect the low sensitivity of this technique. Second, it is well known that many transcripts display extremely poor correlation to the amount of the corresponding protein when assayed at the single cell level. CTSK expression in Fig 4f may reflect the known expression of CTSK in tendon cells and not necessarily indicate the presence of CTSK-lineage periosteal cells in the pool of cells analyzed. The discordance between LEPR and CXCL12 in the Fig 4f analysis is puzzling given that these markers (LEPR+ vs CXCL12+ "CAR" cells) have been largely shown to refer to the same population (e.g., Matsushita et al. Nat Commun 2020). Perhaps the most likely explanation is that the CXCL12 cells analyzed here do not represent the well-studied CXCL12 abundant reticular cells present in marrow stroma that co-express LEPR, but that these cells are other CXCL12+ populations. Thus, without further qualification or supporting studies, the unstated implication here that CXCL12 positivity relates to well studied CXCL12+ mesenchymal populations in bone may be misleading. Overall it is recommended that this analysis be carefully reconsidered and that the less reliable portions of this analysis be either clearly qualified or omitted.

Response: We thank the reviewer for this comment and agree that interpretations of scRNAseq data without supporting flow cytometry analyses should be stated carefully. However, we were very cautious in the revised version of the manuscript to not include any interpretation relative to the similarities between Ctsk- or Cxcl12-positive populations within skeletal muscle and bone. We stated both in the results and discussion that we detected these markers among others in the FAP/MP clusters, but this did not make further statement. However, we did not remove the

data from Supplementary Figure 4f, as these observations were confirmed in other datasets from the literature (see reviewer figure 1 below). In addition, we applied the same cautiousness for the analyses of markers defined by the group of Chan et al. in Supplementary Figure 4e. These analyses did not change our conclusion based on other marker expression confirmed by FACS that “Skeletal muscle contains a heterogeneous population of skeletal muscle mesenchymal progenitors derived from Prx1 mesenchymal lineage, expressing common markers with skeletal stem/progenitor cells from bone and contributing to bone repair.”

To take in consideration this comment by the reviewer, we added the following sentence in the first paragraph of the discussion: “Given the diversity and heterogeneity of mesenchymal progenitors, further analyses will be required to clarify the similarities and functionalities between mesenchymal progenitors located in skeletal muscle and bone.”

Reviewer Figure 1

Reviewer Figure 1: Mesenchymal marker expression across different datasets.
a, UMAP projection of the 4 datasets integrated: whole mononucleated cells from Tabula Muris consortium (grey dots), Giordani L. et al (blue dots),

Hic1CreERT;Rosa^{tdTom} sorted skeletal muscle cells from Wilder Scott R. (red dots) and Prx1Cre;RosamTmG sorted skeletal muscle cells (green dots). b, Unsupervised clusterization of the 4 datasets integrated results into 17 clusters. Cell populations are delimited by a black dotted line. c, Expression of *Ctsk* and *Cxcl12*, and very low expression of *LepR*, *Grem1*, *Nes* and *Mx1* observed in 4 independent datasets.

2. It is recommended that the bone volume in revised Fig 4c be displayed in units corresponding to actual voxel size (e.g., mm³).

Response: We have changed the unit to mm³ in Fig4c and also in Fig7b and Supplementary Fig8d.

3. As the authors note, imatinib has many targets as a fairly nonspecific kinase inhibitor with well known pleiotropic effects. Some acknowledgement of this limitation in the text is needed to balance the statement that it inhibits PDGFRa. Additionally, qualification is needed to acknowledge that imatinib may impact many cell types present in addition to PRX1-lineage muscle resident cells.

Response: We made changes in the text of the last result section and in the discussion to take in account this comment.

4. The bar graph provided in Fig 4c indicating that all of the fractures in both polytrauma and control groups "healed" appears to be at odds with the statement in the text and the left panels of Fig 4c noting an absence of bone bridging in 100% of polytrauma mice. It should be clarified what "healed" means here and how this fits with the non-union/non-bridging seen in polytrauma.

Response: We thank the reviewer for this comment as the bar graph in Fig4c but also in Fig7c and Supplementary Fig8d may cause confusion. In fact, the graph in Fig4c shows that 100% of the fractures healed in control group and 100% did not heal in the polytrauma group. We modified the representation of the graph and removed the color code to avoid this confusion.

Reviewer #2 (Remarks to the Author):

The authors have improved the manuscript substantially and have addressed all my concerns satisfactorily.

Reviewer #3 (Remarks to the Author):

The authors have adequately addressed my concerns.

Just one point of clarification regarding the ablation experiment. The goal here would be to graft the EDL from a mouse in which resident MPs express iDTR (or a TAM-inducible DTA) into a non-transgenic mouse. Once grafted into a recipient mouse, presumably the transplanted MPs within the EDL could be selectively ablated using TAM or the DTA fragment. A *Pdgfra-CreERT2* mouse would work well for this strategy and has been successfully used by others to ablate muscle-associated MPs (Wosczyzna et al., 2019, Cell Reports).

Response: We thank the reviewer for this suggestion. We are indeed exploring this strategy in ongoing projects.